# Soil dielectric characterization during freeze-thaw transitions using L-band coaxial probe and soil moisture probes

Alex Mavrovic[1-2], Renato Pardo Lara[3], Aaron Berg[3], François Demontoux[4], Alain Royer[5-2], Alexandre Roy[1-2]

[1] Université du Québec à Trois-Rivières, Trois-Rivières, Québec, G9A 5H7, Canada
[2] Centre d'Études Nordiques, Université Laval, Québec, Québec, G1V 0A6, Canada
[3] University of Guelph, Guelph, Ontario, N1G 2W1, Canada
[4] Laboratoire de l'Intégration du Matériau au Système, Bordeaux, 33400 Talence, France
[5] Centre d'Applications et de Recherches en Télédétection, Université de Sherbrooke, Sherbrooke, Québec, J1K 2R1, Canada

*Correspondence to:* Alex Mavrovic (Alex.Mavrovic@uqtr.ca)

**Abstract.** Soil microwave permittivity is a crucial parameter in passive microwave retrieval algorithms but remains a challenging variable to measure. To validate and improve satellite microwave data products, precise and reliable estimations of the relative permittivity ($\varepsilon_r = \varepsilon/\varepsilon_0 = \varepsilon'-j\varepsilon''$; unitless) of soils are required, particularly for frozen soils. In this study, permittivity measurements were acquired using two different instruments: the newly designed open-ended coaxial probe (OECP) and the conventional Stevens HydraProbe. Both instruments were used to characterize the permittivity of soil samples undergoing several freeze/thaw cycles in a laboratory environment. The measurements were compared to soil permittivity models. The OECP measured frozen ($\varepsilon'_{frozen} = [3.5;6.0]$, $\varepsilon''_{frozen} = [0.46;1.2]$) and thawed ($\varepsilon'_{thawed} = [6.5;22.8]$, $\varepsilon''_{thawed} = [1.43;5.7]$) soil microwave permittivity. We also demonstrate that cheaper and widespread soil permittivity probes operating at lower frequencies (i.e. Stevens HydraProbe) can be used to estimate microwave permittivity given proper calibration relative to an L-band (1–2 GHz) probe. This study also highlighted the need to improve dielectric soil models, particularly during freeze/thaw transitions. There are still important discrepancies between *in situ* and modelled estimates and no current model accounts for the hysteresis effect shown between freezing and thawing processes which could have a significant impact on freeze/thaw detection from satellites.

**Keywords:** Open-ended coaxial probe, Freeze-thaw cycles, Soil permittivity, Microwave radiometry, Soil emission modelling

## 1 Introduction

The current generation of L-band (1–2 GHz) satellite-based radiometers offers a unique opportunity to monitor soil moisture and freeze/thaw cycles due to its global coverage and revisit time of only a few days (Kerr et al., 2012; Roy et al., 2015; Rautiainen et al., 2016; Colliander et al. 2017; Derksen et al., 2017; Wigneron et al., 2017). These satellites include the European Space Agency Soil Moisture Ocean Salinity mission (SMOS; Kerr et al., 2010), the National Aeronautics and Space Administration (NASA) Soil

Moisture Active Passive mission (SMAP; Entekhabi et al., 2010) and the NASA/CONAE (Comisión Nacional de Actividades Espaciales) joint Aquarius mission (Le Vine et al., 2010). Information about the physical state of the soil is retrieved from microwave observations by using radiative transfer models to simulate the interaction between electromagnetic waves and the surface (Attema and Ulaby, 1978; Mo et al., 1982; Ulaby et al., 1990; Bracaglia et al., 1995; Huang et al., 2017). Such models have already been applied to obtain information on the characteristics of snow cover (Lemmetinen et al., 2016), the state of vegetation (Mo et al., 1982; Rodríguez-Fernández et al., 2018, Fan et al., 2018), soil moisture (Kerr et al., 2012; Mialon et al., 2015; Colliander et al. 2017) and soil freeze/thaw state (Kim et al., 2012; Rautiainen et al., 2016; Derksen et al., 2017; Roy et al., 2017a, 2018 and 2020; Prince et al., 2019).

Permittivities of the landscape constituents are crucial components of the dielectric models used to solve the electromagnetic equations governing the interaction between microwave and surface. The permittivity of a medium ($\varepsilon$, in F/m) determines its behavior when exposed to an electric field. The relative permittivity is the ratio between a medium's permittivity and that of a vacuum ($\varepsilon_r = \varepsilon/\varepsilon_0 = \varepsilon' - i\,\varepsilon''$; unitless; hereafter relative permittivity will stand for permittivity). Permittivity is characterized by a complex number, where the real part ($\varepsilon'$) describes the translation and rotation of molecular dipoles, which drives the wave propagation, and the imaginary part ($\varepsilon''$) describes the energy loss (absorption) associated with this process (Griffiths, 1999). The real and imaginary parts are linked through the Kramers–Kronig relations (Klingshirn, 2012), therefore they are not fully independent. A medium that strongly opposes the application of an external electric field displays a high permittivity (e.g. $\varepsilon'_{water} \approx 78$–$79$ in the 1–2 GHz frequency range; Pavlov and Baloshin, 2015) and a medium that does not strongly oppose an external electric field displays a low permittivity (e.g. $\varepsilon'_{air} \approx 1$).

Because of water's high permittivity, it dominates the microwave signal observed by satellite-based radiometers. Similarly, soil moisture retrieval algorithms exploit the high contrast in water-soil-air permittivity differences. However, the water phase also plays an important role in soil permittivity. When water freezes, the molecules become bound in a crystal lattice and the permittivity drops drastically compared to liquid water (i.e. $\varepsilon'_{ice} \approx 3$). The permittivity drop observable within freezing soils translates into a higher microwave emission from the ground. This allows for the retrieval of the ground state (freeze/thaw) from passive microwave observations (Zuerndorfer et al., 1990; Judge et al., 1997; Zhao et al., 2011; Rautiainen et al. 2012; Roy et al., 2015; Derksen et al, 2017). Soil permittivity is especially important in radiative transfer models since it acts as a boundary condition in the models. As microwave permittivity is challenging to measure in field settings, it is typically derived from empirical relationships and physical properties. Nonetheless, many uncertainties remain in the relationship between soil permittivity and soil physical parameters (Montpetit et al., 2018; Moradizadeh and Saradjian, 2016). This is especially evident during the winter when, in many cases, fixed values are introduced in data analysis algorithms due to a lack of better estimates or, in other cases, data are simply not available during winter. The difficulty in gathering *in situ*

permittivity data at microwave frequencies represents a major hindrance in the parameterization and validation of soil permittivity models, which induces high uncertainties in soil permittivity estimates. This is further complicated by the frequency dependence of permittivity.

Therefore, there is a need to collect better permittivity estimates for the validation of microwave observations and models. However, the majority of instruments deployed to validate microwave permittivity models, such as soil moisture sensors, use measurement frequencies (50–70 MHz) well outside the range of the concerned satellite observations (1400–1427 MHz). Until now, in the absence of a better alternative, the assumption that MHz and L-Band microwave soil permittivity are equivalent has been widely used to validate SMAP and SMOS algorithms (Roy et al., 2017a; Lemmetyinen et al., 2016), although this assumption was never rigorously tested. Furthermore, very few instruments used in field conditions continuously measure microwave permittivity in the frequency range of satellite sensors (Demontoux et al., 2019 and accepted). In addition, only a few laboratory studies have used L-Band permittivity measurements, and most of the available studies have focused on thawed soil samples (Bircher et a., 2016a and 2016b; Demontoux et al., 2017).

The goal of this laboratory-based study is to assess OECP L-band permittivity measurements in frozen soils and the implications of substituting them with permittivity estimates taken at lower frequency by: 1) evaluating the  L-band permittivity of different types of soil in frozen and unfrozen conditions using an open-ended coaxial probe (OECP); 2) comparing the OECP measurements with those from a commercially available soil moisture probes operating at a lower frequency  (i.e. the Stevens HydraProbe) to evaluate the potential of these lower cost probes to estimate L-Band permittivity, and; 3) comparing the soil permittivity measurements captured with both devices against that predicted from soil permittivity models currently used in L-band passive microwave retrieval algorithms. This paper is structured as follows: Section 2.1 describes permittivity instruments used in this study; Section 2.2 gives an overview of two soil permittivity models used for satellite retrieval; Section 3 provides information on the study sites, data collection and laboratory setup. Lastly, in Sections 4 and 5, we compare and contrast the OECP measurements, commercial probe measurements and model simulations.

## 2 Theoretical background

### 2.1 Soil permittivity instruments

This study compares the permittivity estimates from two devices, an OECP and the Stevens HydraProbe, the following sections briefly describe these instruments.

### 2.1.1 Open-Ended Coaxial Probe (OECP)

An OECP was developed by the Université de Sherbrooke (UdeS) and Université du Québec à Trois-Rivières (UQTR) to monitor the permittivity at L-band frequencies of tree trunks (Mavrovic et al., 2018), tree leaves (Holtzman et al., accepted) and snow (Mavrovic et al., 2020) (Fig. 1a). The OECP acts as a coaxial waveguide, and the reflection coefficient at the interface of its open edge and the probed medium is measured by a reflectometer connected to the OECP. This reflectometer acts as both an electromagnetic wave generator and a reflection coefficient measuring instrument for frequencies from 1 to 2 GHz. The reflection coefficient (i.e. magnitude of the reflected and incident electric field ratio) depends on the permittivity of the probed medium. The permittivity is retrieved from the reflection coefficient using a specific calibration based on open (air), short (copper plate) and standard samples (saline solutions of known permittivity) (Filali et al., 2006 and 2008). The permittivity of a wide range of materials can be measured by the OECP as long as it is possible to ensure the probe's open edge makes contact with a flat and smooth surface. This probe has already been described in detail and calibrated on known permittivity surfaces (Mavrovic et al., 2018 and 2020). The sensing depth of the OECP is defined as the maximal depth at which a medium is polarized due to the incident electric field, and as such contributes to the electromagnetic wave reflection. The sensing depth is inversely proportional to the medium's permittivity and proportional to the magnitude of the electric field generated by the reflectometer, which displays a constant power output of 10 dBm (Fig. 1b). The OECP typical sensing depth approaches 1 cm under dry soil conditions and the cylindrical probed volume is about 3.5 cm wide in diameter (Figure 2). Under wet soil conditions, the sensing depth shrinks down to 0.4 cm. Under wet soil conditions, the sensing depth shrinks down to 0.4 cm. The probe system is operational in remote environments since it is easily transportable, sensibly sized (low weight and small dimensions), energy efficient, weatherproof, and operates at low temperatures. The OECP integrates a permittivity measurement in less than a second and does not require destructive sampling, although the user must be careful to avoid air gaps between the probe and the soil. While tested on reference solids, the OECP displays uncertainties under 3.3% for real permittivity and under 2.5% or 0.04 (whichever is greater) for imaginary permittivity (Mavrovic et al., 2018).

### 2.1.2 HydraProbe

The HydraProbe (HP) is a commercial soil moisture probe, from *Stevens Water Monitoring Systems, Inc.,* that uses coaxial impedance dielectric reflectometry to measure soil permittivity (HydraProbes Soil Sensor User Manual, 2018). A digital model of the HP using the SDI-12 protocol was employed. The probe consists of a cylindrical casing which houses the electronics as well as four stainless steel tines (0.3 cm in diameter, 5.7 cm long) that protrude from a metal base plate (4.2 cm diameter). Three tines are arranged in a circle 3.0 cm in diameter around a central tine. The HP operates at 50 MHz and probes a larger volume than the OECP, ranging between approximately 40 and 350 cm$^3$. The HP soil complex permittivity computation is derived from the impedance measurements between the steel tines, which depends mainly on the liquid water content of the soil surrounding the tines (Kraft et al., 1988, Campbell et al., 1988 and 1990, Seyfried et al., 2004). Thus, the HP measures real and imaginary soil permittivities (uncertainties of ± 0.2 or ± 1%, whichever is

greater) as well as temperature (± 0.3°C). From these two variables, soil moisture is estimated using an empirical relationship calibrated for the given soil type (uncertainties between ± 0.01 and 0.03 volumetric water content depending on soil type), with individual calibrations resulting in slightly lower uncertainties (Seyfried et al., 2005; Burns et al., 2014; Rowlandson et al., 2013). This probe is widely used to measure soil moisture for meteorological and agricultural applications. It is deployed along several meteorological station networks (e.g Tetlock et al. 2019). Figure 2 illustrates typical probed volumes for the OECP (dry ~10 cm$^3$, wet ~5 cm$^3$) and HP (dry ~40 cm$^3$, wet ~350 cm$^3$) under dry and wet soil conditions.

## 2.2 Soil permittivity models

Two models commonly used in the remote sensing community for the retrieval of the soil freeze-thaw state were selected.

### 2.2.1 Zhang's Model

The model from Zhang et al. (2010) (henceforth Zhang's model) is a semi-empirical soil model for estimating microwave soil permittivity from soil physical characteristics. It is an extension of the semi-empirical mixing dielectric model (SMDM) adapted to frozen soils from Dobson et al. (1985). Zhang's model is based on dielectric mixing for soil/air/water mixture to estimate soil permittivity at microwave frequencies:

$$\varepsilon^\alpha = f_s \varepsilon_s^\alpha + f_a \varepsilon_a^\alpha + f_{fw} \varepsilon_{fw}^\alpha + f_{bw} \varepsilon_{bw}^\alpha + f_i \varepsilon_i^\alpha \tag{1}$$

where $\varepsilon$ is the permittivity of the overall soil mixture, $\alpha$ a constant shape factor (optimized at 0.65 by Zhang et al., 2003), $f$ the fraction of each component in the soil mixture and the subscripts $s$, $a$, $i$, $fw$ and $bw$ refer respectively to solid soils, air, ice, free water and bound water. The approximation of combining free and bound water is made in the model to avoid evaluating the challenging bound water permittivity ($\varepsilon_w$). Also, air contribution to permittivity is negligible ($\varepsilon_a \approx 1$). Zhang's model evaluates the unfrozen water fraction ($f_w$) in soil near the freezing point in order to obtain a continuous transition between the solid and liquid phases of water. An empirical exponential decay function ($f_w = A \cdot |T_{soil}|^{-B}$) is used to estimate the liquid water fraction in the freezing soils, the ice fraction is determined from the liquid water fraction and the total amount of water in the soil. The parameters A and B of the previous function were empirically estimated based on soil types (Zhang et al., 2003). Solving eq. 1 to obtain an expression for soil mixture permittivity from constant and measurable parameters, Zhang et al. (2010) obtained:

$$\varepsilon^\alpha = 1 + \frac{\rho_b}{\rho_s}(\varepsilon_s'^\alpha - 1) + f_w^\beta \varepsilon_w^\alpha - f_w + f_i \varepsilon_i^\alpha - f_i \tag{2}$$

where $\rho_b$ represents soil bulk density, $\rho_s$ soil specific density and $\beta$ is a parameter that depends on soil composition. The input parameters required by Zhang's model to evaluate all variables in eq. 2 include

frequency (set at 1.4 GHz for this study), soil moisture (main driver for soil permittivity), temperature, dry bulk density and composition (clay, silt and sand fractions) (Zhang et al., 2003 and 2010; Mironov, 2017).

**2.2.2 Temperature Dependable Generalized Refractive Mixing Dielectric Model (TD GRMDM)**

The TD GRMDM is a semi-empirical model that estimates the microwave permittivity of a soil from its
physical properties using a mixing dielectric approach similar to Zhang's model (Mironov et al., 2010). The model accounts for the effect of soil granulometry, temperature and water liquid content through empirical relationships. This model allows for the distinction of bound and free water, giving each of these components a distinct dielectric spectrum. The computational implementation of the TD GRMDM used in this experiment was provided by members of the CESBIO team (Centre d'Etudes Spatiales de la Biosphère, Toulouse, France)
that worked on the operational product of the SMOS mission which used TD GRMDM as one of its modelling components. The input parameters required in TD GRMDM are frequency (set at 1.4 GHz for this study), soil moisture, temperature, dry bulk density and clay fraction. Soil moisture is the main parameter driving soil permittivity. This model was built and validated on a soil database comprising the full range of textures covered by the SMOS mission (Mialon et al., 2015; Mironov et al., 2009 and 2010). However, with respect
to the soil water freeze/thaw state, TD GRMDM is a binary model. All water in the soil is either thawed or frozen, therefore the freeze/thaw transition appears as a discontinuity. The model, however, allows for offsetting the freeze/thaw transition temperature to account for freezing point depression. TD GRMDM uses fixed values for frozen soils with no dependency on temperature, ice fraction or soil composition.

**3 Data and methods**

**3.1 Methods**

Two experiments were performed in this study, the first under fast freeze/thaw transition conditions (one-time air temperature adjustment), and the second under slow transition conditions with small progressive increases in air temperature.

**3.1.1 Fast freeze/thaw transition**

Continuous permittivity measurements were conducted on mineral and organic soil samples going through two or three consecutive freeze/thaw cycles in a NorLake2 mini-room walk-in controlled temperature chamber (5.55 to 19.11 m$^3$ volume) equipped with a CP7L control panel at the School of Environmental Science of the University of Guelph (UofG). The soil samples were previously collected from their respective
study sites (see Sect. 3.2) in PVC or plastic containers. The OBS sample was collected using a rectangular container, while the other samples were collected using cylindrical containers. The containers were placed in an insulated cardboard box (28x38x33 cm for a volume of 3.5x10$^4$ cm$^3$) filled with sand to surround the soil samples (Fig. 3). This setup was intended to simulate the hot/cold front coming from the surface by isolating

the sides and bottom of the soil samples. The OECP and HP were horizontally inserted into undisturbed soil and centered at a depth of 2.5 cm below the soil surface with sufficient spacing between the probes and the soil samples edges to ensure that the probed volumes are restricted to the limits of the soil samples (Fig. 3). Special care was deployed to ensure no air gap was found between the OECP and the undisturbed soil, but without applying extra pressure on the probe. The Fig. 3a and 3b setup discrepancies only reflect the two distinct containers used for soil collection at different sites, both configurations ensured sufficient spacing for undisturbed measurements. OECP measurements were performed at only one position in each experiment because only one OECP was available. The setup of Fig. 3b only includes one HP position because of containers' size limitation. For the organic soil samples, into which multiple probes were inserted, sufficient spacing (~ 7.5 cm with the OECP and > 1 cm between the HP) between probes was ensured to avoid probe interaction. The OECP was calibrated (see Sect. 2.1.1) in the temperature-controlled chamber at +10°C. The OECP can operate at a wide range of temperature and was tested to temperature down to -30°C in the Canadian Arctic (Mavrovic et al., 2020). Beside the OECP, the Planar R54 reflectometer (Copper Mountain Technologies) generating and measuring the electromagnetic waves is graded for [-10 +50] °C temperature range and the Pasternack coaxial cable joining the OECP and the reflectometer for [-50 +205] °C temperature range. The OECP calibration displays a slight temperature dependency, where the calibration drift showed a 0.5% increase in permittivity when using a calibration at -15 °C compare to a calibration at 10 °C. This calibration drift is small compared to the measurement uncertainties (±3.3% for real permittivity and ±2.5% for imaginary permittivity; Mavrovic et al., 2018).

HP output signals were logged with a CR800 datalogger (Campbell Scientific, Inc.). Unlike the HP, the OECP does not record temperature. Therefore, a Campbell Scientific temperature probe (model 107) was placed next to the OECP to measure soil temperature. The air temperature of the cold chamber was set at +10°C for thawing cycles (initial air temperature of -10°C) and -10°C for freezing cycles (initial air temperature of +10°C). These experimental conditions allowed for a complete freeze/thaw cycle in approximately 24 hours and were chosen for practical considerations. However, it should be acknowledged that these conditions represent a relatively rapid transition. Permittivity and temperature measurements were set at one-minute intervals for all instruments.

### 3.1.2 Slow freeze/thaw transition

To investigate the effect of a slower freeze/thaw transition on the temperature amplitude of the hysteresis effect, another experimental setup was created in a Climats EXCAL 1411-HE cold chamber (0.138 m$^3$ volume) at the Laboratoire de l'Intégration du Matériau au Système (Bordeaux, France). Since the soil sample and the Polytetrafluoroethylene (i.e. PTFE or TEFLON) container had smaller volumes, the OECP probe was installed on top of the soil sample with its open end in contact with the soil (Fig. 4). Only OECP permittivity measurements were taken in this experiment since an HP sensor was not available. The objective of this experimental setup was to undergo a slow freeze/thaw transition. Measurements were made to cover

a soil temperature range from -20°C to +11.5°C with a variable soil temperature measurement interval to have a finer curve resolution around freezing point. Permittivity measurements were taken only when the cold chamber air temperature measurements stabilized and the fluctuations between the air and soil temperature were under ± 0.1°C. This method was significantly more time-consuming than the fast transition setup, as a full cycle took several days and required heavy user surveillance.

**3.2 Studied soil types**

Studied soil samples were collected from four different sites and consisted of a single homogenous soil layer (Table 1). Care was taken during transportation to the cold chamber to preserve their original state and leave their structure and moisture content as undisturbed as possible.

The first site was located in the boreal forest at the Old Black Spruce Research Station (OBS). This research facility is in northern Saskatchewan near Canada's boreal forest southern limit and is part of the Boreal Ecosystem Research and Monitoring Sites (BERMS). Its soil is rich in organic matter, displays high soil moisture levels for most of the thawed season (Gower et al., 1997), and is further described in Roy et al. (2020). The samples were collected January 27th, 2018.

The remaining sites were all in agricultural fields with mineral soils in southern Ontario, Canada. Soil samples were collected at the University of Guelph's Elora Research Station (sandy loam; collected late fall 2017) as well as on private farms in Cambridge (loamy sand; collected late fall 2017) and Dunnville (clay loam; collected mid-winter 2018). The soils were selected to be representative of a range of soil textures and complement existing research at the three locations. These samples and their collection process are further described in Pardo Lara et al. (2020) and the data are available at the Federated Research Data Repository through the Polar Data Catalog of metadata (PDC; https://dx.doi.org/10.20383/101.0200).

The soil composition and liquid water content of each sample were analyzed (Table 1). A particle size analysis of the OBS sample was completed at the UdeS using a soil sifting approach to determine the sand fractions and a densitometry technique based on Stokes law (Mériaux, 1953 and 1954) for the clay and silt fractions. The particle sizes of the Dunville, Elora and Cambridge samples were all measured using the hydrometer method (Bouyoucos, 1962). Liquid water content was measured using the drying and weighting technique for all soil samples (O'Kelly, 2004).

**4 Results**

**4.1 Experimental results**

Figures 5 to 8 show the complex permittivity of the four soil samples when undergoing consecutive fast freeze/thaw cycles. Of note, the freeze/thaw transitions were reproducible between cycles using both HP and

OECP sensors. Previous work already shown that the OECP is a reliable instrument to measure a medium's permittivity such as tree trunks (Mavrovic et al., 2018), leaves (Holtzman et al., accepted) and snow (Mavrovic et al., 2020). The OECP displays uncertainties under 3.3% and 2.5% for real and imaginary permittivity respectively when tested on reference materials (Mavrovic et al., 2018). In this study, the repeatability of the OECP measurements through several freeze/thaw cycles can also be seen as an indicator

of the reliability of the experimental setup to measure soil permittivity during freeze/thaw transitions with the OECP and HP. Both thawed soil permittivity from the OECP ($\varepsilon'_{thawed}$ = [6.5;22.8], $\varepsilon''_{thawed}$ = [1.43;5.7]) and HP ($\varepsilon'_{thawed}$ = [6.2;21.7], $\varepsilon''_{thawed}$ = [1.7;10.0]) in Table 2 show a strong correlation between permittivity measurement and volumetric liquid water content as expected. For frozen soils (Table 3), the OECP ($\varepsilon'_{frozen}$ = [3.5;6.0], $\varepsilon''_{frozen}$ = [0.46;1.2]) and HP ($\varepsilon'_{frozen}$ = [2.4;7.0], $\varepsilon''_{frozen}$ = [0.47;2.8]) permittivity measurements

do not seem to display any direct relationship with ice fraction or dry bulk density. Hysteresis effects can be observed between the freezing and thawing cycles in Figs. 5 through 8. Although hysteresis is reported in soil freezing studies, this effect was amplified by the temperature transition speed and differences in the sensing volume for temperature and permittivity observations (Pardo Lara et al., 2020 and in review). Fig. 11 shows a slow freeze/thaw transition displaying a hysteresis effect of diminished amplitude, but still

noticeable. The explanation of the freeze/thaw hysteresis effect is further discussed in sect. 5 to highlight the respective impact of the temperature transition speed and the sensing volume of the temperature measurements versus the permittivity measurements. The HP measurements show trends in agreement with the OECP measurements during freeze/thaw transitions, especially for the real permittivity, although the fully frozen and thawed permittivity values display soil type dependent offsets between the OECP and HP

measurements (Tables 2 and 3). The OECP and HP permittivity measurements, compared in the scatterplot of Fig. 9, are similar for the real part (RMSE = 1.03) but show larger discrepancies for the imaginary part (RMSE = 1.82). Across soil types, no systematic bias between OECP and HP real permittivity were observed, although HP imaginary permittivity measurements tend to be systematically higher than OECP measurements, with the trend being more pronounced at higher imaginary permittivity (i.e. at higher liquid

water content). It was expected that the OECP measured imaginary permittivity would be lower than that of the HP because the dielectric loss due to liquid water is more pronounced at L-band (OECP) than in the MHz frequencies (Mätzler, 1987; Artemov and Volkov, 2014).

In most experiments presented, a short surge in permittivity can be observed right after thawing, followed by

a small drop leading to a convergence to a relatively stable permittivity value associated with a fully thawed soil. Further investigation is needed to see if this short surge could be related to moisture migration toward the thawing front and to water percolation through the soil sample toward the end of the thawing transition. It can also be observed that the freeze/thaw transition measurements are steeper with the OECP than the HP. This is probably due to the HP's larger probed volume. Since the instruments measure an average permittivity

for the whole probed volume, the larger probed volume of the HP records an extended freeze/thaw transition because of the longer time required for the freezing/thawing fronts to penetrate the depth of volume probed.

Since the freezing/thawing front is mostly vertically oriented, it is the difference in probes' sensing diameter that causes the difference in transition steepness.

## 4.2 Model Results

Soil parameters from Table 1 were used to drive the TD GRMDM and Zhang's model. Output from the models is shown in Figs. 5 to 9 and summarized in Tables 2 and 3. There are important discrepancies between the data and the models. The TD GRMDM does not simulate the freeze/thaw transition, resulting in a discontinuity in soil permittivity at the freezing point. Zhang's model estimates the ice fraction for a given sub-freezing temperature, displaying a continuous freeze/thaw transition. Even if amplified by the

experimental setup, the hysteresis effect between the freezing and thawing cycles is not simulated by any model since they do not include the evolution of soil properties in time. The divergence between models and data is more prevalent for the imaginary part of the permittivity than for the real part. Zhang's model seems to systematically underestimate frozen soil permittivity, while the TD GRMDM fixed value approach is closer to the measured permittivity although it does not account, when the soil is frozen, for soil composition

or ice content. Lastly, both models overestimated the soil permittivity of thawed samples with high water content according to the results of this study (Fig. 5), which agrees with results from Bircher et al. (2016b). Further investigation would be required to identify the sources of permittivity overestimation in the models, although it is probable that it comes from the difficulty in uncoupled free and bound water in soil permittivity models. The movement of a fraction of water molecules under the soil surface is hindered by solid soil

particles. Those constrained water molecules are described as bound water. Since their ability to align with an electrical field is reduced, the permittivity of bound water is reduced as well (Jones et al., 2002).

## 5 Discussion

The soil temperature offsets from water freezing point are consistent between the OECP and HP measurements for both the freezing and thawing transitions. The difference is ranging from -1.00 to +0.83

ºC when evaluating the soil temperature offset at maximum transition rate (Tables S1 and S2). The main difference between the permittivity measured at microwave and MHz frequencies appears to be a permittivity offset and the temperature span of the freeze/transition dependent on the soil type. Therefore, based on the offsets seen in Tables 2 and 3 and Fig. 9, a calibration equation between L-band and MHz permittivity can be obtained for a given soil. This would allow for the use of low-cost and widespread instrumentation in the

MHz spectrum, such as the HP, to act as surrogate L-band soil permittivity measurements. This opens up the possibility of studies over large areas through already deployed networks. It should be remembered that MHz permittivity measurements have already been used to test SMAP and SMOS algorithm's permittivity under the assumption that the MHz and L-band permittivity are equivalent (Roy et al., 2017a; Lemmetyinen et al., 2016). As our results showed, MHz and L-band soil permittivity trends are close to each other but not

identical, therefore the previous assumption must be reconsidered because neglecting the frequency dependence of soil permittivity induces a bias in the results.

Ground and satellite-based L-band radiometric measurements are very sensitive to the freezing of the first centimeter of soil (Rowlandson et al., 2018; Roy et al., 2017a, b; Williamson et al., 2018). Therefore, the shallower depth (~ 0.4–1 cm) and smaller volume (~4–10 $cm^3$) probed by the OECP makes it a potentially more suitable instrument to study the freeze/thaw signal observed from L-band radiometers.

The hysteresis effect observed in Figs. 5 to 8 was likely amplified by the experimental setup because of the fast temperature transition speed used. Nonetheless, the hysteresis effect is expected to occur because of the asymmetry between the freezing and thawing processes. The classic Zhang's model only takes into account ice fraction below 0°C, the resulting liquid water fraction should not be interpreted as actual liquid water at temperatures below freezing point but rather as an aggregate of the heterogeneous soil temperature. Figure 10 demonstrates the hysteresis effect simulated by using a modified version of Zhang's model that considers ice fraction above and below 0°C. This ice fraction was prescribed following an exponential function ($\frac{e^{T sol}}{e^{T sol}+1}$) around the freezing point with a ±0.5°C temperature offset for the freezing and thawing cycles. For a proper estimation of ice fraction in soil, the evolution of the soil and boundary conditions should be simulated using more complex models like CLASSIC (Melton at al., 2020).

We further tested the hypothesis that the hysteresis amplitude is correlated with the temperature transition speed using an OBS soil sample with a slower freeze/thaw transition rate. The hysteresis effect displayed in Fig. 11 is still noticeable (< 1°C offset from freezing point) but not as pronounced as in Figs. 5 to 8 (between 2°C and 3°C offset from freezing point). Since the soil permittivity has an important impact on brightness temperature as observed by satellite-based radiometers (Roy et al., 2017a and 2017b; Jonard et al., 2018; Prince et al., 2019;), it is notable that this hysteresis effect around freezing point is not taken into account in current soil models used in microwave satellite retrieval algorithms. The omission of this effect may potentially have an impact on freeze/thaw detection products and their validation. It should be noted that this hysteresis effect is not always observed for *in situ* data due to the instrumental uncertainty not being precise enough to conclusively separate the hysteresis effect *in situ* (e.g. Pardo Lara et al., 2020 and in review). The effect might also be mitigated at the pixel scale of modern satellites because of spatial heterogeneity (Roy et al. 2017b).

Based on our simulations, ice fraction representation in Zhang's model results in a more physically appropriate representation of processes around the freezing point and results in freeze/thaw transitions closer to observations. It should be noted that an ice fraction could be implemented in TD GRMDM as well. To reproduce the hysteresis effect at freeze/thaw transition, two approaches are possible. An empirical approach could be used by implementing a double threshold using distinct ice fraction empirical relationships for 1) the freezing and 2) the thawing cycle. This empirical approach would require determining independently for each transition type the freezing/thawing hysteresis amplitude as a temperature offset between the state

transition and 0°C. This would depend on liquid water content, textural composition, solute concentration, and the pore pressure of the soil (Daanen et al., 2011). The alternative would be to couple dielectric models with soil physical models that integrate the time evolution of soil physical properties (e.g. CLASSIC model; Melton at al., 2020). Soil physical models provide an estimate of the ice fraction through time, which is used by dielectric models to estimate soil permittivity. Such coupling should only impact the freeze/thaw transition where ice fraction is a relevant parameter.

## 6 Conclusion

This study presents soil microwave permittivity measurements during freeze/thaw transitions in the same frequency range as the SMAP and SMOS satellites, as well as future L-band satellite missions. The permittivity measurements were taken using a novel open-ended coaxial probe (OECP). It is shown that lower frequency (MHz) soil permittivity probes can be used to estimate microwave permittivity given proper calibration relative to an L-band probe, which holds significant potential considering the already widespread operational networks of low frequency soil permittivity probes deployed to measure soil moisture. This study also highlighted the need to improve dielectric soil models, particularly during freeze/thaw transitions. We observed noticeable discrepancies between *in situ* data and model estimates, and no current model accounts for the hysteresis effect shown between freezing and thawing processes. Although this phenomenon should be considered as an aggregate of soil temperature heterogeneity rather than actual conditions, it is of relevant interest to study and understand it for all macroscopic to satellite scale applications. Few studies have investigated this hysteresis effect, which could have a significant impact on freeze/thaw detection from satellites. Future work will look to improve soil thermal regime retrieval near the freezing point using permittivity measurements, which is impactful on the evaluation of the carbon budgets of northern regions.

## 7 Acknowledgments

This work was made possible thanks to the contributions of the Canadian Space Agency (CSA), Natural Sciences and Engineering Research Council of Canada (NSERC), Canada Foundation for Innovation (CFI). The experiments in Bordeaux were supported by the Samuel-De-Champlain France-Québec collaborative project (Fonds québécois de la recherche sur la nature et les technologies, FQRNT). We would also like to thank Bilal Filali, PhD, for his contribution to the design and manufacture of the probe, along with Simone Bircher, Arnaud Mialon, Yann Kerr and the entire CESBIO team in Toulouse for their contributions to probe testing and collaboration with Bordeaux laboratory. A special thanks to Jean-Pierre Wigneron from INRA for providing the codes to run the TD GRMDM model. Finally, we would like to recognize the reviewers who improved the paper by their useful comments.

## 8 Competing interests

The authors declare that they have no conflict of interest.

## 9 Data availability

The research data can be accessed by direct request to the author.

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

 **Figures**

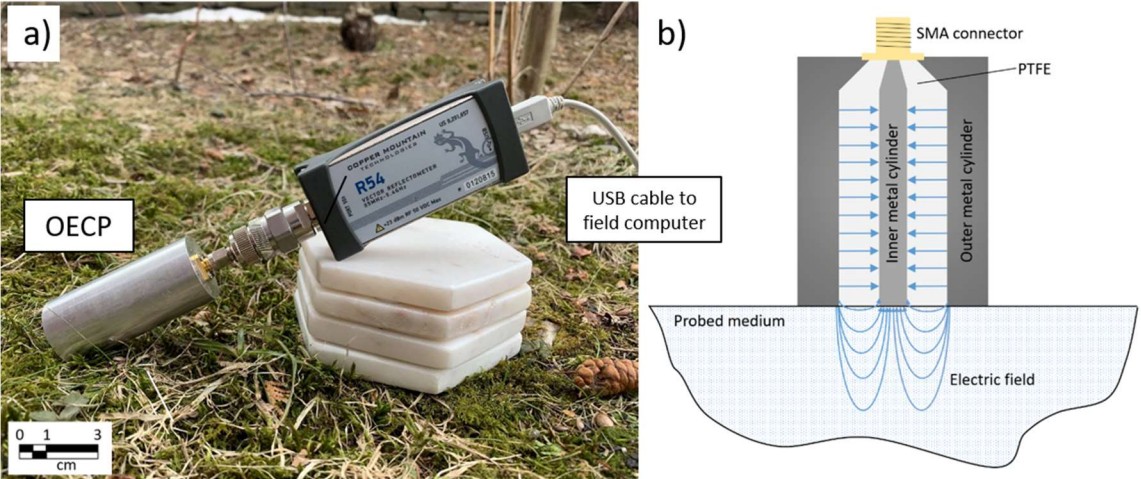

**Figure 1: (a) OECP for permittivity measurement. The control program provided by the Planar R54 reflectometer manufacturer is operated with a field computer. The probe is connected to the Planar R54 reflectometer using a SMA/N cable or adaptor. (b) Diagram of the electrical field produced by the OECP.**


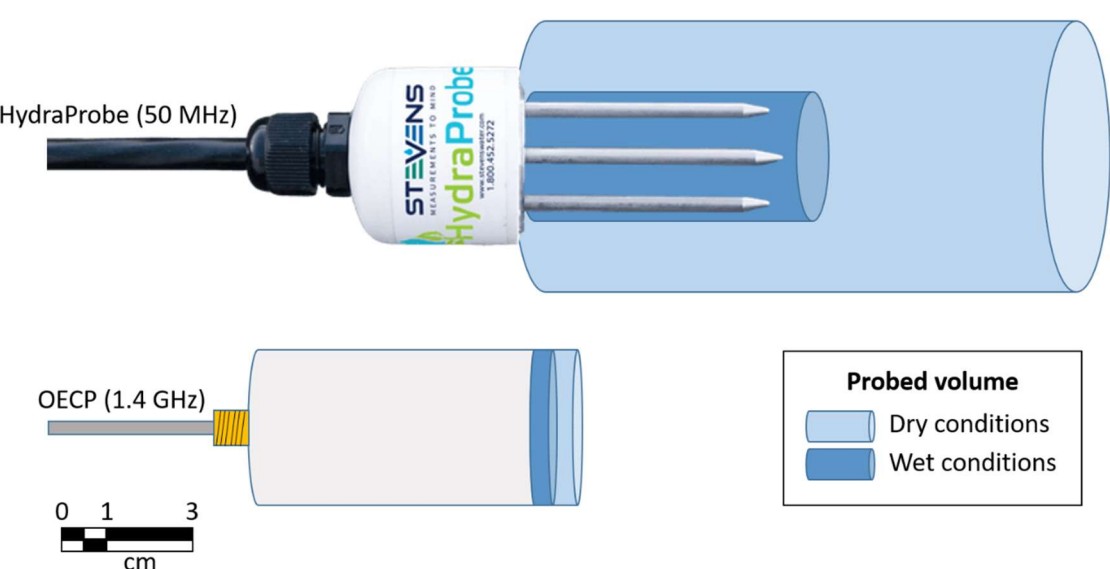

**Figure 2: Approximate probed volume (blue) of the HydraProbe (top) and OECP (bottom) for relatively dry and**
**wet soil conditions. The probed volume is also influenced by soil type.**

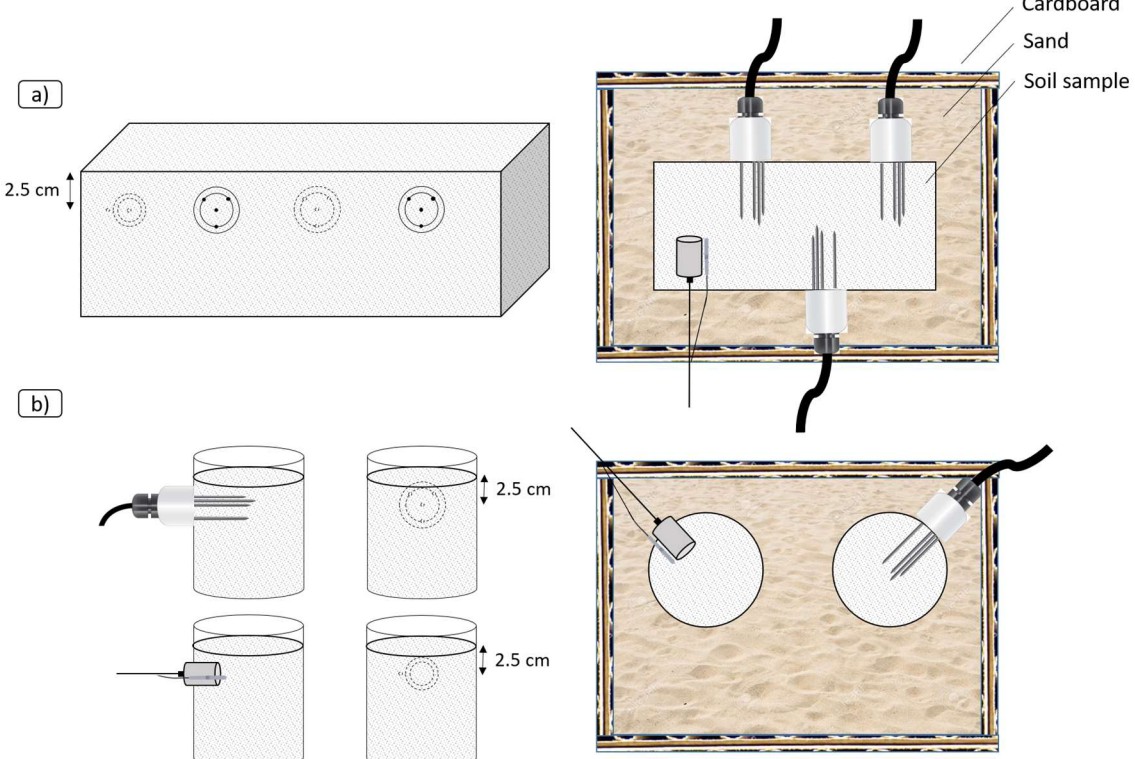

**Figure 3: Top view of the cold chamber experimental setup at Guelph University (Ontario, Canada) for fast transition experiment. Setup for (a) the OBS sample, 11x24x12cm for a volume of $3.2 \times 10^3$ cm$^3$ and (b) the Ontario samples, height of 12 cm and diameter of 10 cm for a volume of $9.4 \times 10^2$ cm$^3$.**


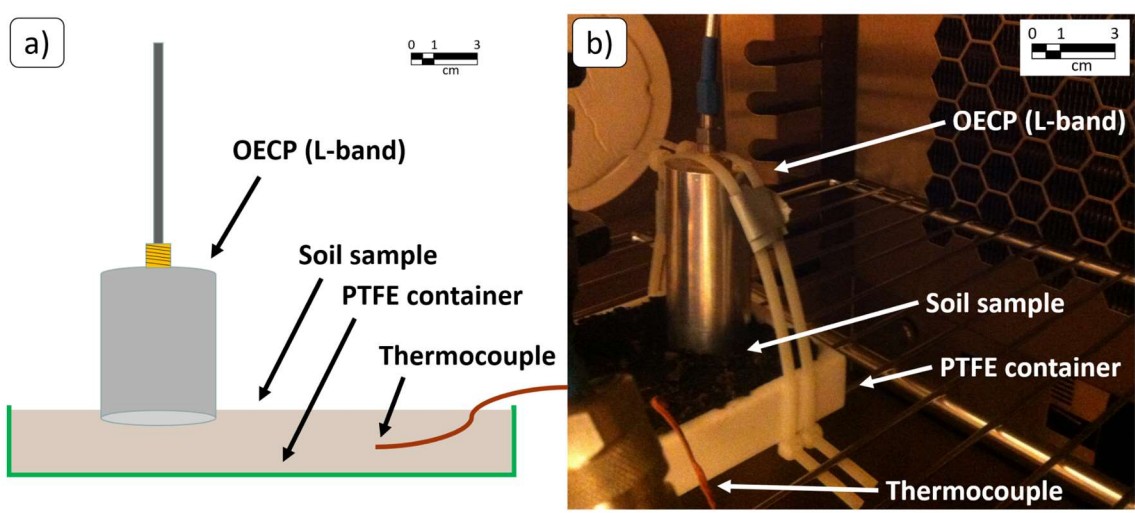

**Figure 4: (a) Side view and (b) photo of the cold chamber experimental setup at the Laboratoire de l'Intégration du Matériau au Système (Bordeaux, France) for the slow freeze/thaw transition.**


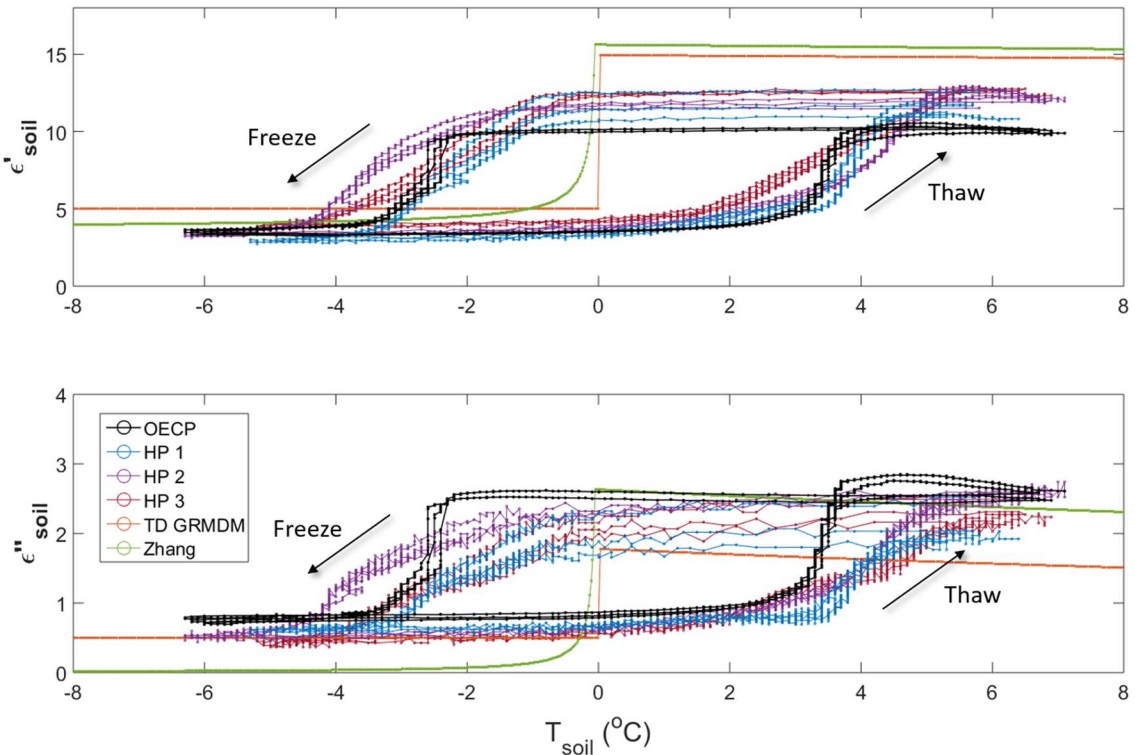

**Figure 5: Real (ε') and imaginary (ε'') permittivity of an organic soil sample from the Old Black Spruce site (see Table 1) during freeze/thaw cycles in a cold chamber environment. The OECP and HP instruments monitored soil permittivity, where TD GRMDM and Zhang are model results. The hysteresis effect displayed here is amplified by the experimental setup (discussed in the text). Experiment conducted from February 1st to February 7th, 2018.**

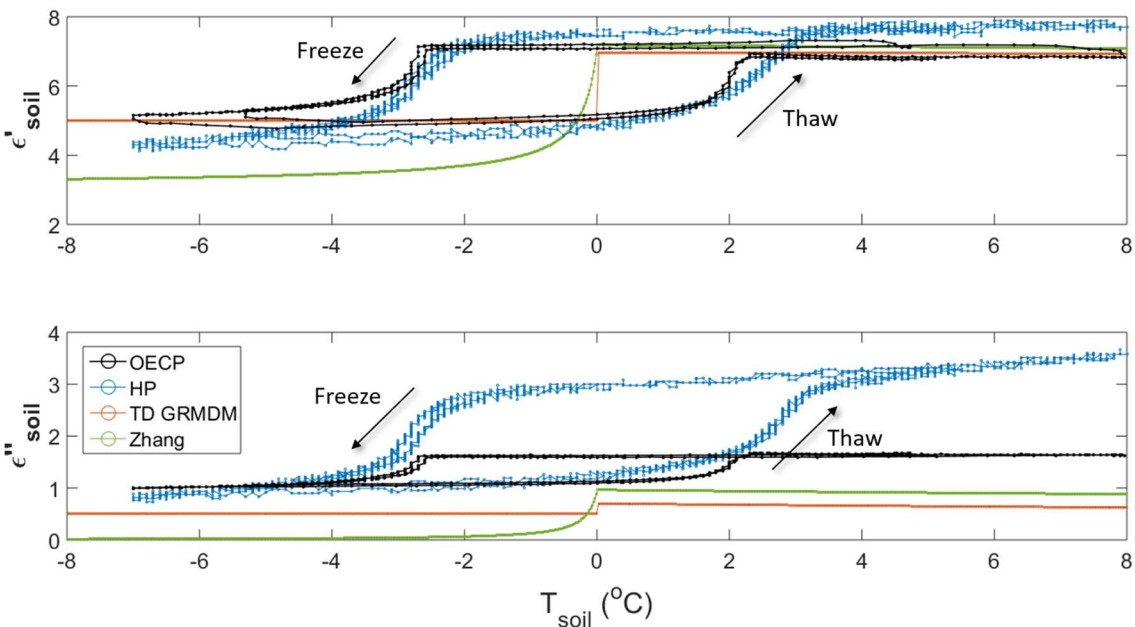

**Figure 6: Same as Fig. 3 but for the sandy loam soil sample (see Table 1). Experiment conducted from April 15th to April 19th, 2018.**

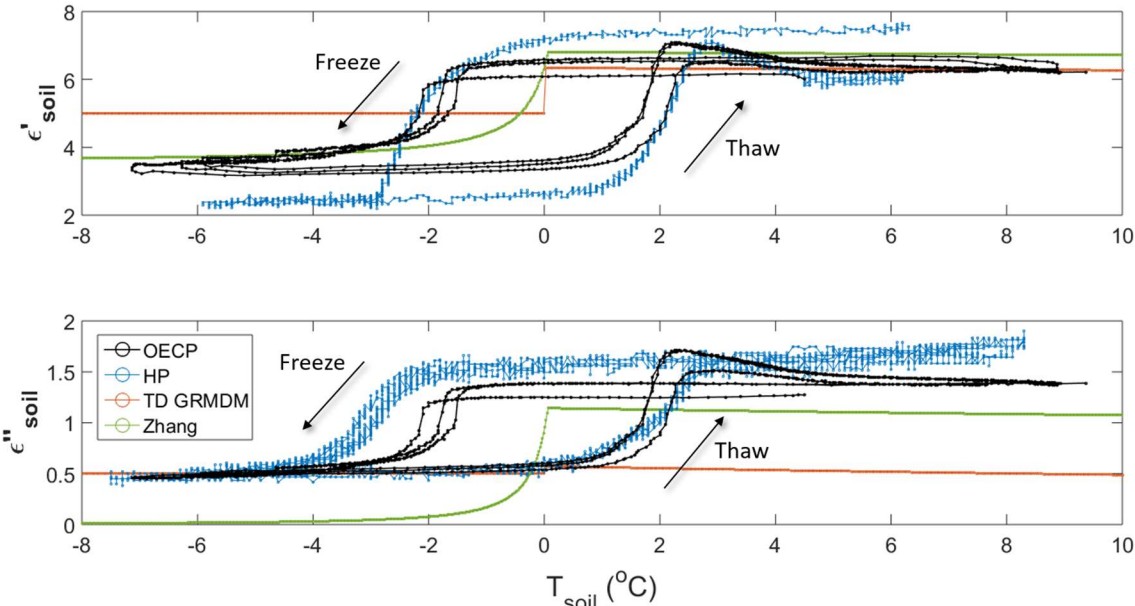


**Figure 7: Same as Fig. 3 but for the loamy sand soil sample (see Table 1). Experiment conducted from March 29th to April 6th, 2018.**

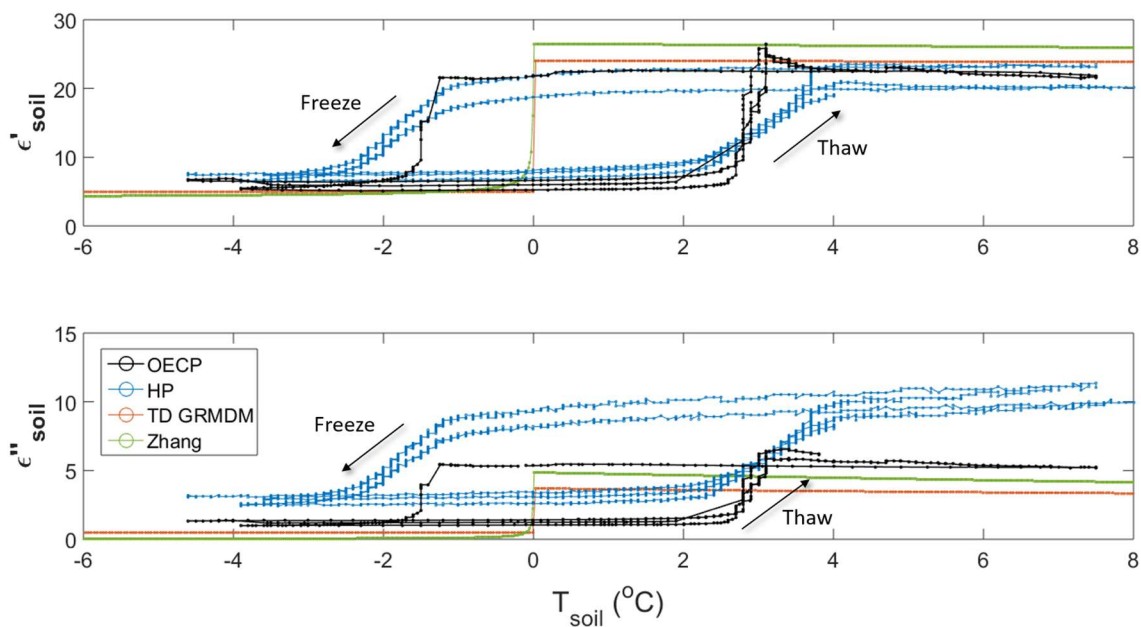


**Figure 8: Same as Fig. 3 but for the clay loam soil sample (see Table 1). Experiment conducted from April 6th to April 15th, 2018.**

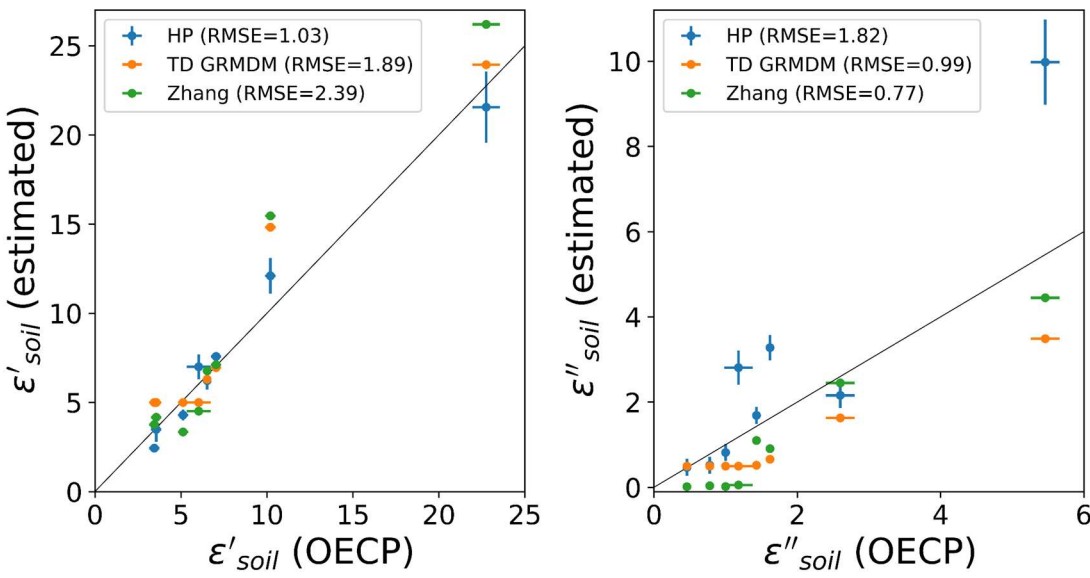

**Figure 9: OECP real (ε') and imaginary (ε'') permittivity compared to HP (instrument), TD GRMDM (model) and Zhang (model) with the OECP as the reference. The black line is the 1:1 reference ratio and the root-mean square error is given in parentheses (RMSE).**

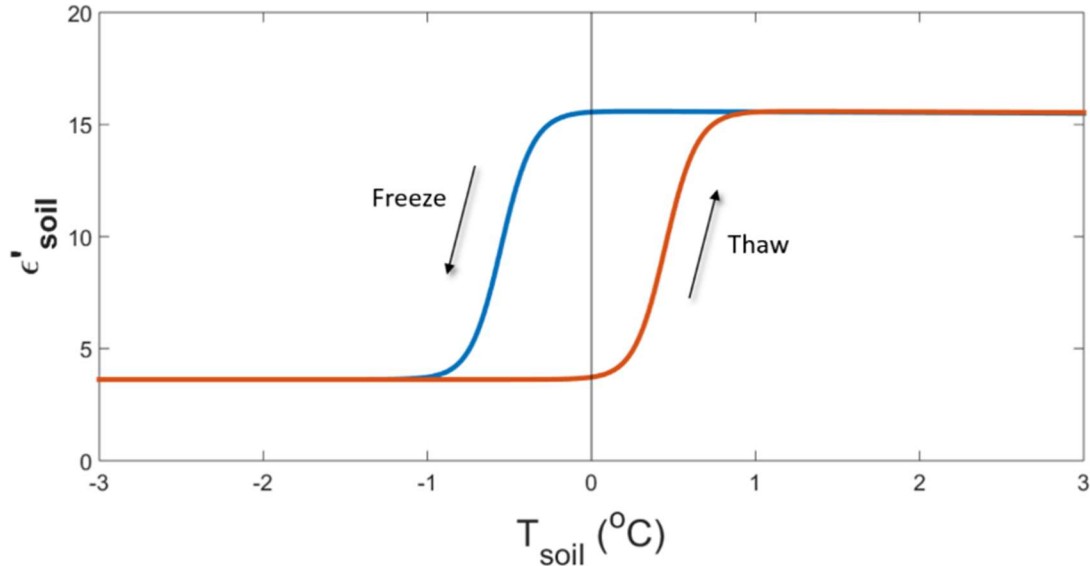

**Figure 10: Expected hysteresis effect between freeze and thaw cycles. This theoretical curve was produced using an adapted version of Zhang's model and the soil composition of the OBS sample.**

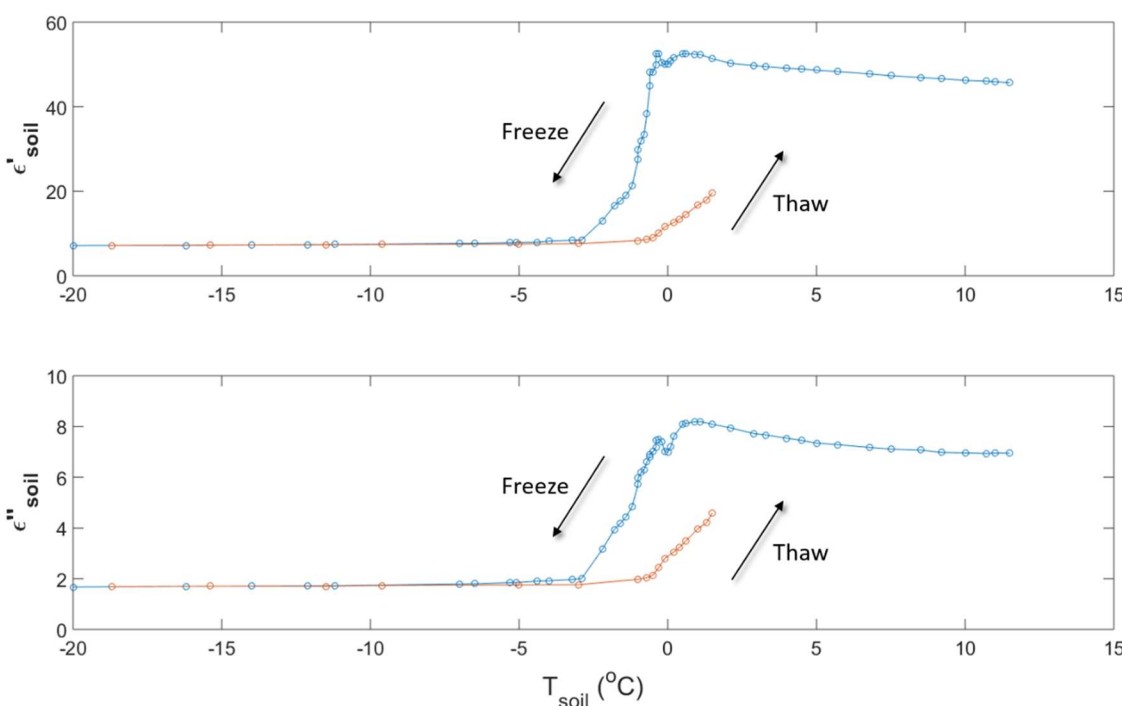

**Figure 11: Real (ε') and imaginary (ε'') permittivity of an organic soil sample from the Old Black Spruce site (collected May 3rd, 2017) during a slow freeze/thaw cycle in a temperature-controlled chamber environment. Experiment conducted July 12th, 2017.**







**Tables**

**Table 1: Soil composition and physical properties. The Old Black Spruce site is located in the boreal forest in Saskatchewan, Canada, and the three other sites are located in agricultural fields in southern Ontario, Canada. $f_i(V)$ and $f_i(G)$ stands for volumetric and gravimetric liquid water content, respectively. $\rho_d$ stands for dry bulk density.**

| Soil type | Site | Lattitude/ Longitude | Gravimetric composition | | | | Physical properties | | |
| --- | --- | --- | --- | --- | --- | --- | --- | --- | --- |
| | | | Organic % | Clay % | Sand % | Silt % | $f_i(V)$ m³/m³ | $f_i(G)$ kg/kg | $\rho_d$ kg/m³ |
| Organic | Old Black Spruce | 53º59' N 105º07' W | 59 | 2.36 | 29.85 | 8.79 | 0.30 | 0.83 | 356.2 |
| Sandy Loam | Elora | 43º39' N 80º25' W | N/A | 10 | 54 | 36 | 0.115 | 0.079 | 1450 |
| Loamy Sand | Cambridge | 46º26' N 80º20' W | N/A | 2.5 | 78.4 | 19.1 | 0.068 | 0.038 | 1780 |
| Clay Loam | Dunville | 42º52' N 79º44' W | N/A | 28 | 33 | 39 | 0.42 | 0.30 | 1400 |

**Table 2: Modelled and measured complex permittivity of thawed soils. The permittivity in the 5°C to 6°C temperature range (stable plateau) is averaged over the multiple freeze/thaw cycles depicted in Figs. 5 through 8. Absolute and relative uncertainties (in parentheses) are based on instrument precision and measurement variability.**

| Soil type | $\varepsilon'_{thawed\ soil}$ | | | | $\varepsilon''_{thawed\ soil}$ | | | |
| --- | --- | --- | --- | --- | --- | --- | --- | --- |
| | OECP | HP | TD GRMDM | Zhang | OECP | HP | TD GRMDM | Zhang |
| Organic | 10.2 (±0.3/2.9%) | 12.1 (±1.0/8.3%) | 14.83 | 15.46 | 2.6 (±0.2/7.7%) | 2.2 (±0.3/13.6%) | 1.63 | 2.45 |
| Sandy Loam | 7.0 (±0.3/4.3%) | 7.6 (±0.2/2.6%) | 6.95 | 7.12 | 1.62 (±0.04/2.5%) | 3.3 (±0.3/9.1%) | 0.66 | 0.91 |
| Loamy Sand | 6.5 (±0.2/3.1%) | 6.2 (±0.5/8.1%) | 6.30 | 6.77 | 1.43 (±0.05/3.5%) | 1.7 (±0.2/11.8%) | 0.52 | 1.10 |
| Clay Loam | 22.8 (±0.8/3.5%) | 21.7 (±2.0/9.2%) | 23.94 | 26.20 | 5.7 (±0.2/3.5%) | 10.0 (±1.0/10.0%) | 3.49 | 4.45 |

**Table 3: Same as Table 2 but for frozen conditions (-5° to -6°C).**

| Soil type | $\varepsilon'_{frozen\ soil}$ | | | | $\varepsilon''_{frozen\ soil}$ | | | |
| --- | --- | --- | --- | --- | --- | --- | --- | --- |
| | OECP | HP | TD GRMDM | Zhang | OECP | HP | TD GRMDM | Zhang |
| Organic | 3.6 (±0.3/8.3%) | 3.5 (±0.7/20.0%) | 5 | 4.17 | 0.78 (±0.04/5.1%) | 0.5 (±0.2/40.0%) | 0.5 | 0.039 |
| Sandy Loam | 5.1 (±0.3/5.9%) | 4.3 (±0.3/7.0%) | 5 | 3.35 | 1.00 (±0.04/4.0%) | 0.8 (±0.2/25.0%) | 0.5 | 0.020 |
| Loamy Sand | 3.5 (±0.3/8.6%) | 2.4 (±0.2/8.3%) | 5 | 3.76 | 0.46 (±0.04/8.7%) | 0.47 (±0.2/42.6%) | 0.5 | 0.017 |
| Clay Loam | 6.0 (±0.7/11.7%) | 7.0 (±0.7/10.0%) | 5 | 4.51 | 1.2 (±0.2/16.7%) | 2.8 (±0.4/14.3%) | 0.5 | 0.055 |