# Peer review of "Soil dielectric characterization during freeze-thaw transitions using L-band coaxial probe and soil moisture probes"

_Hydrology and Earth System Sciences, 2020_

## Referee Comment (RC1) · Jan Hofste (Referee) · 28 Jul 2020

By Jan G. Hofste, University of Twente, The Netherlands.

Date: 2020 07 28

—-General Comments:—-

This manuscript describes a laboratory experiment in which the relative dielectric permittivity at L-band of a variety of different types of soil are measured during the freeze <-> thaw process with a dedicated microwave open-ended coaxial probe (OECP) and a commercial soil moisture probe: Hydraprobe (HP). The measurements with both

the OECP and HP show a clear hysteresis effect associated with the freeze and thaw process in the graphs of complex relative dielectric permittivity (epsilon) versus soil temperature. Although there are some differences in the measured epsilon between the OECP and HP the authors argue that soil moisture sensors such as HP, which are relatively cheap, tested and verified, can be used construct validation networks for passive microwave remote sensing. Additionally, the manuscript addresses that current models for soil epsilon don't incorporate the freeze <-> thaw hysteresis effect.

The experiment, its results, and proposed application in building validation networks for soil dielectric permittivity with soil moisture probes, I consider as a valuable contribution to the microwave remote sensing community. The title does not fully reflect the contents of the paper, I think you could add "with a soil moisture probe" in the end. Description of the experimental design should be improved. Also the description and explanation of the observed freeze <-> thaw hysteresis should be more elaborate. Finally, throughout the paper the structure of the sentences can be improved.

—-Specific Comments:—-

[1] Line 188 and Figure 3: "The OECP and HP were fully ..." Why was the OECP inside the OBS soil sample and not inside the other three soil samples? And if it was buried inside the OBS sample would not this disturb the sample? I suppose the sample structure was better preserved in the configuration of figure 3b.

[2] General remark on the samples and measurement setup. With the OBS sample HP measurements were taken at three positions. As Figure 5 shows the measured responses at these three positions varies. Why were there not also measurements at multiple positions for the OECP with the OBS sample? And why were the other 3 samples not also measured with the HP (and the OECP) at multiple positions? Was this because the OBS sample was expected to be less homogeneous due to the organic content? And why only one sample per soil type was measured? The choices the authors made in this regard should be explained in the text, even if simply for practical

reasons.

[3] Line 243: The amplification of the hysteresis -effect by the setup, is it possible to explain this in the text with a few sentences? You refer to this hysteresis amplification later on, it would be better if the reader could find an explanation for this effect in this manuscript rather than somewhere else (the reference). You can of course leave the reference.

[4] Lines 252 - 255: I am not sure whether I completely agree with your explanation. You ascribe the difference in measured transition to the different probing volumes of both sensors and that with the HP there is a longer time required for the freezing/thawing front to penetrate the probed volume. The way I see it for both sensors the temperature gradient is from top/bottom (because of your nice trick of placing sand around the sample) so ideally the progression of this freezing/thawing front is the same for both sensors. What is different is the diameter of the sensor's probing volume, see Figure 2. For the OECP this is roughly half the diameter of the HP, which would explain a more abrupt transition. Another difference is the length of the probing volume. For wet conditions the length for the HP is about 15x that of the OECP. The way I see it the HP then performs 15 OECP measurements at 15 different positions. If there is variation in the soil over this length, the HP then shows a kind of average transition over all these 15 volumes. This is the reason I was also wondering why the OECP was not used at multiple positions within one sample (comment [2]). You could maybe test your volume explanation by taking the OECP result and scale it to the HP volume.

[5] Figures 5 - 8: During the thawing process there appears to be maximum epsilon' and epsilon'' directly after the main thawing process after which the epsilon' and epsilon'' decrease again slightly. This is effect is most pronounced in Figure 7. Do you have an explanation for this effect?

[6] Lines 270 - 272: Based on the Figures 5 - 8 I find the freeze/thaw transitions not similar. Can the differences of the OECP and HP measurements be explained by the

difference in probing volume? Also you mention that the main difference between the OECP and HP measurements are the epsilon values at the end of the cycle, at the "stable plateaus" as you call it. But isn't the hysteresis just as important? Perhaps if a found calibration equation for a given soil is applied to the HP results the freeze/thaw hysteresis is more like that of the OECP?

[7] Lines 283 - 289: What do the authors want to say with this paragraph? Is the point that, should a network of (tried, tested, and cheap) hydraprobes be installed over a large area as surrogate L-band permittivity sensors (lines 274 – 275), one must realize that the volume over which it measures is not exactly what radiometers probe?

[8] Line 300: What hypothesis do the authors refer to?

[9] Figure 11: With the freezing cycle you see both epsilons increase first before they decrease rapidly when all soil freezes. Why don't we see this behaviour during of the freezing fast freeze/thaw experiment? We do see it during the thawing (see comment [5]), are these processes linked?

—-Technical corrections:—-

[10] Line 20: You state that you show in the manuscript that the OECP is a suitable device for measuring epsilon …. The demonstration that OECP can measure the epsilon of any homogeneous material is given in your previous studies Mavrovic2018 and Mavrovic2020, not in this manuscript. In this manuscript you use the OECP to quantify the performance of the HP. I propose to change the sentence to .. the OECP measured the frozen soil epsilon' to be 3.5 to 6.0, the epsilon" to be 0.4 – 1.2 etc.

[11] Line 41, 42: Cite not only papers that use the tau-omega model for microwave scattering of vegetation. Give examples of papers that solve the radiative transfer equations differently, such as the Tor Vergata model (Bracaglia, Ferrazzoli, and Guerriero, RSE, 1995) or the MIMICS model (Ulaby, Sarabandi, et al., 1990 IJRS).

[12] Line 50: Propose to change to: "Permittivity is characterized by a complex number,

where the real part describes the translation and rotation of molecular dipoles, which drive the wave propagation, and the imaginary part describes the energy loss associated with this process." Further I propose to refer to a textbook on electrodynamics, for example: Griffiths D.J., Introduction to Electrodynamics.

[13] Line 63: remove the word passive here.

[14] Line 75: "to collect permittivity estimates", propose to change to "to collect better permittivity estimates." Also ". . . for the validation of passive microwave instruments". Propose to change to ".. for the validation of microwave radiometric observations". Or something similar, but it's the observations than need to be validated.

[15] Line 86: Add OECP between assess and L-band.

[16] Line 109: The reflectometer generates an electromagnetic wave, not only a propagating electric field.

[17] Line 110: Over what frequency band were the measurements performed?

[18] Line 116: "The penetration depth of the . . ." This sentence is too vague for my taste. I propose something like: "The sensing depth of the OECP is the maximum depth at which the medium is polarized due to the incident electric field, and as such contributes to the reflection of the EM wave backwards into the coax."

[19] Line 118: "The magnitude of this effective electric. . ." the effective electric field has not been defined or explained previously. I assume you refer the resulting electric field in the medium? Which is the sum of the original electric field coming from the coax E0, which polarizes (rotates and or translates) the molecules and the electric field produced by the rotated or displaced molecules themselves Ed. Latter counters E0, which counters Ed, which counters E0 etc. You end up with a resulting electric field E, which is actually lower in magnitude for a higher epsilon.

[20] Line 119: You describe the electric field generated by the reflectometer in terms of power (dBm = 1 mW) which is incorrect. I propose to state simply that the generated

power is 10 dBm.

[21] Line 131: If applicable, note what type of Hydraprobe you used (for example type A or B100 or . . .)

[22] Line 137: ".. it uses the ratio of the incident and reflected waves to numerically solve Maxwell's equations, yielding the impedance and complex permittivity." That the device solves the Maxwell's equations sounds far-fetched to me. One of the papers on found on the Hydraprobe website should provide you with a better (quick) description on how the device works. In my understanding the Hydraprobe indeed works similar to the OECP: The epsilon of the material between the steel tines determines the characteristic impedance (symbol Z0 typically, or its inverse the characteristic admittance Y0) . The reflection of the The steel tines, together with the material (soil) they are in, forms a microwave transmission line with characteristic impedance Z0 (or its inverse Y0). The reflection coefficient, measured by the device, is dependent on this Z0.

[23] Line 140: mention $\pm 0.01$ and $\pm 0.03$ are uncertainties.

[24] Line 185: It confused me whether the samples were collected from the temperature chamber or from the sites. It is the latter I understand? Further, I propose to use distinguishing names. Call the PVC boxes with the collected soil 'samples' as is, but refer to the cardboard boxes, filled with samples and surrounding sand, with a different name. Maybe sample assembly. Indicate these names in Figure 3. This way you can mention for example that the "sample assemblies were placed in the temperature chamber and were subjected to 3 (? mention this as well) freeze/thaw cycles".

[25] Lines 243 – 246. Authors state that trends of OECP and HP are "very similar" and the fully frozen/thawed epsilon values are "also similar". I disagree with this description. Judging from Figures 5 - 8 there are significant differences. These differences and explanations for their causes are discussed further down in the text.

[26] line 251: ".. the freeze/thaw transition measurements are smoother with the HP

than..." Perhaps there is a better alternative for "smoother", perhaps "less abrupt"? Also sentence should be "We also observe that the measured freeze/thaw transitions are less abrupt (?) with the measurements of the HP than with the OECP." Same for line 261.

[27] Lines 325 - 327: The question whether the OECP correctly measures the epsilon in not shown in this manuscript. It is implied by your earlier work, see also comment [10].

[28] Figures 5 – 8: To make comparison easier I propose to let all figures have the same axis limits for epsilon' and epsilon", even if this implies having only one figure per page. Further I would recommend using more contrasting colours for the curves and to plot the graphs in vector format (PDF).

[29] Tables 2 and 3: Besides the absolute uncertainty also indicate the relative uncertainty.

––––––––––––––––––––––––––

---

## Referee Comment (RC2) · Anonymous Referee #2 · 28 Jul 2020

General comments:

The manuscript by Mavrovic et al. conducted permittivity measurements of different soil types with various soil water content using OECP and HydraProbe at frequency of L-band and 50MHz, respectively. Two experiments, fast freeze/thaw transition and slow freeze/thaw transition, were designed. Two soil dielectric model, TD GRMDM and Zhang's model, were driven by the known inputs to simulate the real and imaginary part of soil permittivity. By comparing permittivity measurements between OECP and HydraProbe during freeze/thaw cycles, they demonstrated there are differences of permittivity characteristic between L-band and MHz instruments and suggested the

necessities to make proper calibration. By comparing the permittivity measurements and model simulations, they reported the observable discrepancies and highlighted the need for soil dielectric models to take into account the hysteresis effect. Such work is under the research topic to evaluate satellite microwave data products from the in situ permittivity measurements (MHz frequency).

The topic of this manuscript is of interest to the readers of HESS and the measurements can be potentially of importance to the microwave related researches. However, in its current form, the uncertainties regarding the measurements are not detail, which make it hard to judge the validity of the comparison of OECP and HydraProbe measurements. The difference between OECP and HydraProbe measurements is not only from the frequency dependence of permittivity, but also can come from the fact that they are not measuring the same volume of soil samples. As the temperature range of this experiment is large, the temperature dependence of OECP and HydraProbe measurements matters. In addition, the presentation of results is with inaccuracies and can be further explained. Given the current form of the manuscript, I cannot recommend its publication. I expect it suitable for publication in HESS with convinced presentation of measurements and results. Please see below my specific comments.

Specific comments:

Title, Abstract: I can not see any details about the description of soil dielectric characterization in the Abstract. Please consider either adjust the title or adding the relevant text in Abstract.

1 Introduction

Line 71: "The high uncertainties in soil permittivity models result from the difficulty in gathering in situ permittivity..." as from my understanding, the uncertainties in soil permittivity models can come from the parameters is not well defined by the in situ permittivity measurements. please clarify this sentence.

2 Theoretical background

Line 104 & 130: Section numbers are incorrect.

Line 124: please explain the temperature dependence of OECP measurements. As OECP undergoes a large variation of temperature (e.g., -10°C to 10°C), how does OECP perform under such conditions? At which temperature OECP is calibrated? Please make a clarification.

Sect. 2.2 please consider presenting the equations used for TD GRMDM and Zhang's model, maybe can put in the appendix. As later you proposed a modification of Zhang's model to consider the hysteresis effect, It is better to present the equations and clear introduce how you make modifications.

Line 193: what is HPP?

3 Data and methods

Sect. 3.1.2 Slow freeze/thaw transition Please explain the purpose for this experiment. Please describe the temperature settings and add information about the measuring interval of OECP and HP measurements.

3.2 Studied soil types

Maybe I have misunderstandings here. How many soil samples were collected and then used in this experiment? Are these soil samples for each site with the same moisture content?

Line 221: When is the experiment conducted?

4 Results

In this section, Figures 5-8 are presented. While only a general description was presented. Lacking of the characteristic of soil dielectric, the difference among Figures 5-8, the difference between fast and slow freeze/thaw transition measurements.

Line 242: please explain "Although hysteresis should be expected because of the latent heat of fusion of water".

Line 246: "with offsets depending on the soil type" please consider presenting the results more detail.

Line 267-268: "both models overestimated the soil permittivity of thawed samples with high water content according to the results of this study." please explain such overestimation.

5 Discussion

Line 295: please consider presenting the equations of the modified version of Zhang's model.

Line 296: "consider ice fraction above 0°C" is the artefact or the real conditions? Please make explanations.

Line 300: please specify what is "the hypothesis".

Line 316: please explain how you implement a double "threshold"

6 Conclusions

In the current form, conclusion appears not informative compare to the Abstract. Please consider making modifications, adding more information.

Technical comments:

Line 95: considering change into "Section 2.2 gives an overview of two soil permittivity models"

Figure 4: please add the plotting scale to indicate the dimensions.

Figure 10: how is it reproduced? Please indicate the equations, the used parameters.

Figure 11: where are (a) and (b) on the figures?

Tables: Please consider using the consistent format

---

## Referee Comment (RC3) · Anonymous Referee #3 · 27 Aug 2020

Review of the manuscript "Soil dielectric characterization at L-band microwave frequencies during freeze-thaw transitions" by Mavrovic et al.

The manuscript presents interesting measurements of soil permittivity at L-band during the freeze-thaw cycles. Results are compared with two commonly used models (Mironov's model and Zhang's model) and hysteresis effects are observed especially for the fast freeze/thaw transitions. The reviewer found these experimental results are valuable and suggests to accept it for publication after addressing the following concerns.

Lines 63-64: Not only for L-band, higher frequencies are also able to retrieve the landscape freeze/thaw state. (e.g. Zuerndorfer et al., 1990; Judge et al., 1997; and Zhao et al., 2011). And if possible, future measurements could be extended to higher frequencies, which is important to retrieve snow properties and soil properties under the snow. Please refer to: Zuerndorfer, B. W., England, A. W., Dobson, M. C., & Ulaby, F. T. (1990). Mapping freeze/thaw boundaries with SMMR data. Agricultural and Forest Meteorology, 52(1-2), 199-225. Judge, J., Galantowicz, J. F., England, A. W., & Dahl, P. (1997). Freeze/thaw classification for prairie soils using SSM/I radiobrightnesses. IEEE Transactions on Geoscience and Remote Sensing, 35(4), 827-832. Zhao, T., Zhang, L., Jiang, L., Zhao, S., Chai, L., & Jin, R. (2011). A new soil freeze/thaw discriminant algorithm using AMSR‐E passive microwave imagery. Hydrological Processes, 25(11), 1704-1716.

Lines 125: Would it cause uncertainties of measurement when applying different pressures to the soil with the OECP probe?

Lines 248: How are the data points selected for Figure 9, as there are many measurements as shown from Figure 5 to 8. The challenge is how to well model the soil permittivity during the freeze-thaw transitions, and data points during the freezing/thawing period should be included.

Figure 9: please specify those numbers are for RMSE in the figure.

Lines 290: It is very interesting that the hysteresis effects were observed during the permittivity measurement. As mentioned below by the authors, an empirical approach could be used by implementing a double threshold. It is suggested to do so to discuss the improvement of the model performance compared with results from Figure 9.

---

## Author Comment (AC1) · 30 Oct 2020

**Revision of Manuscript: "Soil dielectric characterization during freeze-thaw transitions using L-band coaxial probe and soil moisture probes" by Alex Mavrovic** *et al.*

In blue: Reviewer's comments.
R= Reviewer; P = Page; L = Line as they appear in the Original manuscript version
G= General comment followed by numbering

In black: Answers to referees.
P=Page; L=Line; Track change version
*In black and italic: Modification added to text.*

**Comments from the Reviewers:**

**Reviewer #1:**

Synopsis:

This manuscript describes a laboratory experiment in which the relative dielectric permittivity at L-band of a variety of different types of soil are measured during the freeze <-> thaw process with a dedicated microwave open-ended coaxial probe (OECP) and a commercial soil moisture probe: Hydraprobe (HP). The measurements with both the OECP and HP show a clear hysteresis effect associated with the freeze and thaw process in the graphs of complex relative dielectric permittivity (epsilon) versus soil temperature. Although there are some differences in the measured epsilon between the OECP and HP the authors argue that soil moisture sensors such as HP, which are relatively cheap, tested and verified, can be used construct validation networks for passive microwave remote sensing. Additionally, the manuscript addresses that current models for soil epsilon don't incorporate the freeze <-> thaw hysteresis effect.

The experiment, its results, and proposed application in building validation networks for soil dielectric permittivity with soil moisture probes, I consider as a valuable contribution to the microwave remote sensing community. The title does not fully reflect the contents of the paper, I think you could add "with a soil moisture probe" in the end. Description of the experimental design should be improved. Also the description and explanation of the observed freeze <-> thaw hysteresis should be more elaborate. Finally, throughout the paper the structure of the sentences can be improved.

We made substantial improvement to the manuscript by adding more details on the experimental design and the models' equations and assumptions. The explanations of the freeze/thaw hysteresis effect were also expended to highlight the impact of the temperature transition speed and the temperature sensing volume versus the permittivity sensing volume on the hysteresis amplitude. Finally, the title was adjusted to better reflect the objectives of the study.

Specific comments:

[1] R1, P6, L188 and Fig. 3: "The OECP and HP were fully… " Why was the OECP inside the OBS soil sample and not inside the other three soil samples? And if it was buried inside the OBS sample would not this disturb the sample? I suppose the sample structure was better preserved in the configuration of figure 3b.

The OECP was buried in the OBS soil sample by digging a small-scale trench, alike soil moisture probes are generally installed in the field. The horizontal position of the OECP fully places the probed volume in the undisturbed soil. The OECP was fully buried in the OBS soil sample to simplify the installation. Since the other three samples were smaller, the setup requires that part of the OECP stands out of the container to assure the probed volume stays within the soil samples. Details were added to this paragraph about the experimental setup.

P6-7, L212-214: The OECP and HP were *horizontally inserted into undisturbed soil* and centered at a depth of 2.5 cm below the soil surface *with sufficient spacing between the probes and the soil samples edges to ensure that the probed volumes are restricted to the limits of the soil samples* (Fig. 3).

[2] R1, G1: General remark on the samples and measurement setup. With the OBS sample HP measurements were taken at three positions. As Figure 5 shows the measured responses at these three positions varies. Why were there not also measurements at multiple positions for the OECP with the OBS sample? And why were the other 3 samples not also measured with the HP (and the OECP) at multiple positions? Was this because the OBS sample was expected to be less homogeneous due to the organic content? And why only one sample per soil type was measured? The choices the authors made in this regard should be explained in the text, even if simply for practical reasons.

The OECP is a promising instrument currently developed by the Université de Trois-Rivières and Université de Sherbrooke. Only one OECP was available for the experiment. Logistics is the primary reason for the difference in setup between the OBS soil samples and the others, the soil sample collections were not made in the same type of container for all sites. Even if the cylinder samples are smaller in size, the probed were properly positioned to ensure reliable measurements (see previous response to comment R1[1]). The repeatability of the measurements gives us confidence that the experimental protocol is robust. The explanation for the two distinct setups was added.

P7, L216-218: *The Fig. 3a and 3b setup discrepancies only reflect the two distinct containers used for soil collection at different sites, both configurations ensured sufficient spacing for undisturbed measurements.*

[3] R1, P7, L243: The amplification of the hysteresis -effect by the setup, is it possible to explain this in the text with a few sentences? You refer to this hysteresis amplification later on, it would be better if the reader could find an explanation for this effect in this manuscript rather than somewhere else (the reference). You can of course leave the reference.

Further explanations and reference were added.

P8-9, L286-291: Hysteresis effects can be observed between the freezing and thawing cycles in Figs. 5 through 8, *i.e. a different behavior of permittivity variation depending on whether the ground freezes or thaws*. Although hysteresis *is reported in soil freezing studies, this effect was amplified by the temperature transition speed and differences in the sensing volume for temperature and permittivity observations (Pardo Lara et al., 2020)*. Fig. 11 shows a slow freeze/thaw transition displaying a hysteresis effect of diminished amplitude, but still noticeable.

P9-10, L321-323: Even if amplified by the experimental setup, the hysteresis effect between the freezing and thawing cycles is not simulated by any model since *they do not include the evolution of soil properties in time*.

[4] R1, P8, L252-255: I am not sure whether I completely agree with your explanation. You ascribe the difference in measured transition to the different probing volumes of both sensors and that with the HP there is a longer time required for the freezing/thawing front to penetrate the probed volume. The way I see it for both sensors the temperature gradient is from top/bottom (because of your nice trick of placing sand around the sample) so ideally the progression of this freezing/thawing front is the same for both sensors. What is different is the diameter of the sensor's probing volume, see Figure 2. For the OECP this is roughly half the diameter of the HP, which would explain a more abrupt transition. Another difference is the length of the probing volume. For wet conditions the length for the HP is about 15x that of the OECP. The way I see it the HP then performs 15 OECP measurements at 15 different positions. If there is variation in the soil over this length, the HP then shows a kind of average transition over all these 15 volumes. This is the reason I was also wondering why the OECP was not used at multiple positions within one sample (comment [2]). You could maybe test your volume explanation by taking the OECP result and scale it to the HP volume.

We agree that the freezing/thawing front is mostly vertically oriented and therefore is the probes' volume diameter difference that is mostly responsible for the difference in transition steepness. Clarifications about the point were added. The suggested experiment of using several OECP at different positions is a relevant one, but as mentioned previously, only one probe was available for the experiment, while removing the probe to measure at different depths would have changed the soil properties. Comparing the average of several OECP measurements to the HP measurement would possibly give further insights on the OECP and HP comparability.

P9, L311-315: Since the instruments measure an average permittivity for the whole probed volume, *the* larger probed volume *of the HP* record an extended freeze/thaw transition because of the longer time required for the freezing/thawing fronts to penetrate the depth of volume probed. *Since the freezing/thawing front is mostly vertically oriented, it is the difference in probes' sensing diameter that causes the difference in transition steepness.*

[5] R1, Fig. 5-8: During the thawing process there appears to be maximum epsilon' and epsilon'' directly after the main thawing process after which the epsilon' and epsilon'' decrease again slightly. This is effect is most pronounced in Figure 7. Do you have an explanation for this effect?

It is suspected that this effect is due to water percolation during thawing, but further investigation is needed to confirm this phenomenon. This hypothesis was added to the discussion.

P9, L306-309: *In most experiments presented, a short surge in permittivity can be observed right after thawing, followed by a small drop leading to a convergence to a relatively stable permittivity value associated with a fully thawed soil. Further investigation is needed to see if this short surge could be related to moisture migration toward the thawing front and to water percolation through the soil sample toward the end of the thawing transition.*

[6] R1, P8, L270-272: Based on the Figures 5 - 8 I find the freeze/thaw transitions not similar. Can the differences of the OECP and HP measurements be explained by the difference in probing volume? Also you mention that the main difference between the OECP and HP measurements are the epsilon values at the end of the cycle, at the "stable plateaus" as you call it. But isn't the hysteresis just as important? Perhaps if a found calibration equation for a given soil is applied to the HP results the freeze/thaw hysteresis is more like that of the OECP?

It is correct that the difference in freeze/thaw transition steepness could be explained by the difference in probing volume. The authors share the same point of view that this is probably the main explanation and it is put forward in the Experimental Results section (4.1).

The fully frozen/thawed values comparison between measurements and models consist of the strongest differences observed in this study. The hysteresis is of equal importance, but the trends are similar between the permittivity measurements. This is to say that the hysteresis effect occurs at very similar temperatures.

It is typical to use soil specific calibration equation to produce soil moisture estimates from HP raw permittivity measurements. However, the HP instrument does not allow for customized calibration equation to compute permittivity from raw reflection coefficients.

P9, L 310-315: It can also be observed that the freeze/thaw transition measurements are *steeper* with the *OECP* than the HP. This is probably due to the HP's larger probed volume. Since the instruments measure an average permittivity for the whole probed volume, a larger probed volume will record a *more extended* freeze/thaw transition because of the longer time required for the freezing/thawing fronts to penetrate the depth of volume probed. *Since the freezing/thawing front is mostly vertically oriented, it is the difference in probes' sensing diameter that causes the difference in transition steepness.*

[7] R1, P8, L283-289: What do the authors want to say with this paragraph? Is the point that, should a network of (tried, tested, and cheap) hydraprobes be installed over a large area as surrogate L-band permittivity sensors (lines 274 – 275), one must realize that the volume over which it measures is not exactly what radiometers probe?

The potential in using already deployed HydraProbes networks is laid out in the previous paragraph. The message of this paragraph is that in the case of freeze/thaw algorithms testing, previous studies have shown that the L-band radiometric signal is sensitive to the freezing of the first centimeter of soil which implies that the OECP would be a more suitable instrument to study that phenomenon due to its inherent smaller sensing volume. The paragraph was reshaped to get this point across more clearly.

P10, L348-351: *Ground- and satellite-based L-band radiometric measurements are very sensitive to the freezing of the first centimeter of soil (Rowlandson et al., 2018; Roy et al., 2017a,b; Williamson et al., 2018). Therefore, the shallower depth (~ 0.4–1 cm) and smaller volume (~4–10 cm$^3$) probed by the OECP makes it a potentially more suitable instrument to study the freeze/thaw signal observed from L-band radiometers.*

[8] R1, P9, L300: What hypothesis do the authors refer to?

The hypothesis referred is the one proposed in the previous paragraph about the correlation between the hysteresis effect and the temperature transition speed. It was clarified to avoid further confusion.

P11, L362-363: *We further tested the hypothesis that the hysteresis effect is correlated with the temperature transition speed* using an OBS soil sample using a slower freeze/thaw transition rate.

[9] R1, Fig. 11: With the freezing cycle you see both epsilons increase first before they decrease rapidly when all soil freezes. Why don't we see this behaviour during of the freezing fast freeze/thaw experiment? We do see it during the thawing (see comment [5]), are these processes linked?

It is hypothesized that water displacement inside the sample during freezing is responsible for this surge, similar to the explanation of the surge in Fig. 5-8 (see response to comment R1[5]).

[10] R1, P1, L20: You state that you show in the manuscript that the OECP is a suitable device for measuring epsilon… . The demonstration that OECP can measure the epsilon of any homogeneous material is given in your previous studies Mavrovic2018 and Mavrovic2020, not in this manuscript. In this manuscript you use the OECP to quantify the performance of the HP. I propose to change the sentence to .. the OECP measured the frozen soil epsilon' to be 3.5 to 6.0, the epsilon" to be 0.4 – 1.2 etc.

Suggestion applied and text modified.

P1, L21: *The OECP measured the* frozen…

[11] R1, P2, L41-42: Cite not only papers that use the tau-omega model for microwave scattering of vegetation. Give examples of papers that solve the radiative transfer equations differently, such as the Tor Vergata model (Bracaglia, Ferrazzoli, and Guerriero, RSE, 1995) or the MIMICS model (Ulaby, Sarabandi, et al., 1990 IJRS).

References to radiative transfer models added.

P2, L39-42: Information about the physical state of the soil is retrieved from microwave observations by using radiative transfer models to simulate the interaction between electromagnetic waves and the surface (*Attema and Ulaby, 1978; Mo et al., 1982; Ulaby et al., 1990; Bracaglia et al., 1995; Huang et al., 2017*).

P12, L418-419: *Attema, E., and Ulaby, F.: Vegetation modeled as a water cloud. Radio Sci., 13(2), 357-364, doi:10.1029/RS013i002p00357, 1978.*

P13, L429-430: *Bracaglia, M., Ferrazzoli, P., and Guerriero, L.: A fully polarimetric multiple scattering model for crops, Remote Sens. of Environ., 54(3), 170–179, doi:10.1016/0034-4257(95)00151-4, 1995.*

P17, L664-665: *Ulaby, F., Sarabandi, K., McDonald, K., Whitt, M., and Dobson, M.: Michigan microwave canopy scattering model, Int. J. Remote Sens., 11(7), 1223–1253. doi:10.1080/01431169008955090, 1990.*

[12] R1, P2, L50: Propose to change to: "Permittivity is characterized by a complex number, where the real part describes the translation and rotation of molecular dipoles, which drive the wave propagation, and the imaginary part describes the energy loss associated with this process." Further I propose to refer to a textbook on electrodynamics, for example: Griffiths D.J., Introduction to Electrodynamics.

Suggestion implemented. The Griffiths is a well-known reference to the author, it was added.

P2, L52-54: Permittivity is characterized by a complex number, where the real part ($\varepsilon$') describes the *translation and rotation* of molecular dipoles, which drives *the* wave propagation, and the imaginary part ($\varepsilon$'') describes the *energy loss (absorption) associated with this process (Griffiths, 1999)*.

P14, L501-502: *Griffiths, Introduction to electrodynamics - Third Edition, Pearson, Upper Saddle River, New Jersey, 576 pp., 1999.*

[13] R1, P2, L63: remove the word passive here.

``passive`` removed.

P2, L65-66: The permittivity drop observable within freezing soils translates into a higher microwave emission from the ground.

[14] R1, P3, L75: "to collect permittivity estimates", propose to change to "to collect better permittivity estimates." Also "… for the validation of passive microwave instruments". Propose to change to ".. for the validation of microwave radiometric observations". Or something similar, but it's the observations than need to be validated.

Suggestion implemented.

P3, L79-80: Therefore, there is a need to collect *better* permittivity estimates for the validation of *microwave observations and models*.

[15] R1, P3, L86: Add OECP between assess and L-band.

Added.

P3, L91: The goal of this laboratory-based study is to assess *OECP* L-band permittivity measurements…

[16] R1, P4, L109: The reflectometer generates an electromagnetic wave, not only a propagating electric field.

Clarified.

P4, L113-114: This reflectometer acts as both *an electromagnetic wave generator and* a reflection coefficient measuring instrument *for frequencies from 1 to 2 GHz*.

[17] R1, P4, L110: Over what frequency band were the measurements performed?

Specified.

P4, L113-114: This reflectometer acts as both *an electromagnetic wave generator and* a reflection coefficient measuring instrument *for frequencies from 1 to 2 GHz*.

[18] R1, P4, L116: "The penetration depth of the… " This sentence is too vague for my taste. I propose something like: "The sensing depth of the OECP is the maximum depth at which the medium is polarized due to the incident electric field, and as such contributes to the reflection of the EM wave backwards into the coax."

Sentence reworked.

P4, L120-127: The *sensing* depth of the OECP *is* defined as the maximal depth at which a medium is *polarized due to the incident electric field, and as such contributes to the electromagnetic wave reflection. The sensing depth is proportional to the medium's permittivity and the magnitude of the electric field generated by the reflectometer*, which

*displays a constant power output of 10 dBm* (Fig. 1b). The OECP typical *sensing* depth approaches 1 cm under dry soil conditions and the cylindrical probed volume is about 3.5 cm wide in diameter (Figure 2). Under wet soil conditions, the *sensing* depth shrinks down to 0.4 cm.

[19] R1, P4, L118: "The magnitude of this effective electric… " the effective electric field has not been defined or explained previously. I assume you refer the resulting electric field in the medium? Which is the sum of the original electric field coming from the coax E0, which polarizes (rotates and or translates) the molecules and the electric field produced by the rotated or displaced molecules themselves Ed. Latter counters E0, which counters Ed, which counters E0 etc. You end up with a resulting electric field E, which is actually lower in magnitude for a higher epsilon.

The effective electrical field refers here to the extent of the electrical field influencing the reflection coefficient measurements. The use of this term here seems confusing and not necessary. Therefore, it was removed and the sensing depth was directly referenced.

P4, L122-124: *The sensing depth is proportional to the medium's permittivity and the magnitude of the electric field generated by the reflectometer*, which *displays a constant power output of 10 dBm* (Fig. 1b).

[20] R1, P4, L119: You describe the electric field generated by the reflectometer in terms of power (dBm = 1 mW) which is incorrect. I propose to state simply that the generated power is 10 dBm.

Clarified.

P4, L122-124: *The sensing depth is proportional to the medium's permittivity and the magnitude of the electric field generated by the reflectometer*, which *displays a constant power output of 10 dBm* (Fig. 1b).

[21] R1, P4, L131: If applicable, note what type of Hydraprobe you used (for example type A or B100 or …)

Precisions were added on the HydraProbe used for this experiment.

P4, L137: *A digital model of the HP using the SDI-12 protocol was employed.*

[22] R1, P4, L137: ".. it uses the ratio of the incident and reflected waves to numerically solve Maxwell's equations, yielding the impedance and complex permittivity." That the device solves the Maxwell's equations sounds far-fetched to me. One of the papers on found on the Hydraprobe website should provide you with a better (quick) description on how the device works. In my understanding the Hydraprobe indeed works similar to the OECP: The epsilon of the material between the steel tines determines the characteristic impedance (symbol Z0 typically, or its inverse the characteristic admittance Y0) . The reflection of the The steel tines, together with the material (soil) they are in, forms a

microwave transmission line with characteristic impedance Z0 (or its inverse Y0). The reflection coefficient, measured by the device, is dependent on this Z0.

The HydraProbe Soil Sensor User Manual (2018) section 4.4.1 (p.34) on the theory of operation cites more comprehensive references on how the numerical solution of Maxwell's equations is obtained to derive the complex permittivity from impedance measurements. Those references were added.

P4, L141-143: *The HP soil complex permittivity computation is derived from the impedance measurements between the steel tines, which depends mainly on the liquid water content of the soil surrounding the tines (Campbell, 1990; Seyfried and Murdock, 2004).*

P13, L439-440: *Campbell, J.: Dielectric Properties and Influence of Conductivity in Soils at One to Fifty Megahertz, Soil Sci. Soc. Am. J., 54(2), 332-341, doi:10.2136/sssaj1990.03615995005400020006x, 1990.*

P17, L654-655: *Seyfried, M., and Murdock., M.: Measurement of Soil Water Content with a 50-MHz Soil Dielectric Sensor, Soil Sci. Soc. Am. J., 68(2), 394-403, doi:10.2136/sssaj2004.3940, 2004.*

[23] R1, P4, L140: mention ±0.01 and ±0.03 are uncertainties.

Added.

P4-5, L144-147: Thus, the HP measures real and imaginary soil permittivities (*uncertainties of* ± 0.2 or ± 1%, whichever is greater) as well as temperature (± 0.3°C). From these two variables, soil moisture is estimated using an empirical relationship calibrated for the given soil type (*uncertainties* between ± 0.01 and 0.03 volumetric water content depending on soil type), with…

[24] R1, P6, L185: It confused me whether the samples were collected from the temperature chamber or from the sites. It is the latter I understand? Further, I propose to use distinguishing names. Call the PVC boxes with the collected soil 'samples' as is, but refer to the cardboard boxes, filled with samples and surrounding sand, with a different name. Maybe sample assembly. Indicate these names in Figure 3. This way you can mention for example that the "sample assemblies were placed in the temperature chamber and were subjected to 3 (? mention this as well) freeze/thaw cycles".

Clarifications were added about the distinction between the collection of the samples at the study sites and the freeze/thaw cycles experiment in the temperature-controlled chamber. The components of the sample assembly were identified in Fig. 3. The soil samples underwent 2 or 3 freeze/thaw cycles, which is now specified.

P6, L204-206: Continuous permittivity measurements were conducted on the mineral and organic soil samples going through *two or three* consecutive freeze/thaw cycles in a NorLake2 mini-room walk-in controlled temperature-controlled chamber…

P6, L207-209: The soil samples *were previously collected from their respective study sites (see Sect. 3.2)* in PVC or plastic containers. *The OBS sample was collected using a rectangular container, while the other samples were collected using cylindrical containers.*

Fig. 3: Updated in manuscript.

[25] R1, P7, L243 – 246. Authors state that trends of OECP and HP are "very similar" and the fully frozen/thawed epsilon values are "also similar". I disagree with this description. Judging from Figures 5 - 8 there are significant differences. These differences and explanations for their causes are discussed further down in the text.

The similarity mentioned here was meant to point at the closer similarities between OECP and HP measurements than the model estimates. Since the model results are discussed in the next section, the sentence was reformulated to remove this mention.

P9, L294-296: The HP measurements show trends *in agreement with that of* the OECP measurements during freeze/thaw transitions, especially for the real permittivity*, although the fully* frozen and thawed permittivity values *display soil type dependent offsets* between the OECP and HP measurements (Tables 2 and 3).

[26] R1, P7, L251: ".. the freeze/thaw transition measurements are smoother with the HP than… " Perhaps there is a better alternative for "smoother", perhaps "less abrupt"? Also sentence should be "We also observe that the measured freeze/thaw transitions are less abrupt (?) with the measurements of the HP than with the OECP." Same for line 261.

The term ''smooth'' was replaced through the manuscript by the terms ''steeper'', ''continuous'' and ''extended'' depending on the context.

P5, L167-169: Zhang's model evaluates the unfrozen water fraction ($f_w$) in soil near the freezing point in order to obtain a continuous transition between the solid and liquid phases of water.

P9, L310: It can also be observed that the freeze/thaw transition measurements are *steeper* with the *OECP* than the *HP*.

P9, L311-313: Since the instruments measure an average permittivity for the whole probed volume, a larger probed volume will record a *more extended* freeze/thaw transition because of the longer time required for the freezing/thawing fronts to penetrate the depth of volume probed.

P9, L320-321: Zhang's model estimates the ice fraction for a given sub-freezing temperature, displaying a *continuous* freeze/thaw transition.

P11, L374-376: Based on our simulations, ice fraction representation in Zhang's model results in a more physically appropriate representation of processes around the freezing point and results in *freeze/thaw* transitions *closer to observations*.

[27] R1, P10, L325 - 327: The question whether the OECP correctly measures the epsilon in not shown in this manuscript. It is implied by your earlier work, see also comment [10].

The reliability of OECP measurements are not thoughtfully investigated in this study, although confidence in the reliability of the measurements can be inferred from the repeatability through freeze/thaw cycles. The conclusion was adapted to shift focus from OECP reliability to soil permittivity results and a hint to OECP measurements repeatability was added in the Results section.

P8, L281-282: *The repeatability of the OECP measurements can also be seen as an indicator of the reliability of the measurements.*

P11, L387-392: *This study presents soil microwave permittivity measurements during freeze/thaw transitions in* the same frequency range as the SMAP and SMOS satellites, as well as future L-band satellite missions. *The permittivity measurements were taken using a novel open-ended coaxial probe (OECP).* It is shown that lower frequency (MHz) soil permittivity probes can be used to estimate microwave permittivity given proper calibration relative to an L-band probe*, which holds significant potential considering the already widespread operational networks of low frequency soil permittivity probes deployed to measure soil moisture.*

[28] R1, Fig. 5 – 8: To make comparison easier I propose to let all figures have the same axis limits for epsilon' and epsilon", even if this implies having only one figure per page. Further I would recommend using more contrasting colours for the curves and to plot the graphs in vector format (PDF).

The color palette of figures 5 to 8 was changed to increase color contrast. The authors would prefer to stay with variable y-axis since the data from figure 7 would be hardly visible if put at the same scale of figure 8 (i.e. it would occupy a third of the graph for in the real permittivity graph and a sixth in the imaginary permittivity graph).

Figures 5-8: Updated in manuscript.

[29] R1, Tables 2 and 3: Besides the absolute uncertainty also indicate the relative uncertainty.

Relative uncertainties were added aside absolute values in Table 2 and 3.

[revised manuscript text omitted]
 | 6.0 (±0.7/11.7%) | 7.0 (±0.7/10.0%) | 5 | 4.51 | 1.2 (±0.2/16.7%) | 2.8 (±0.4/14.3%) | 0.5 | 0.055 |

---

## Author Comment (AC2) · 30 Oct 2020

**Revision of Manuscript: "Soil dielectric characterization during freeze-thaw transitions using L-band coaxial probe and soil moisture probes" by Alex Mavrovic *et al.***

In blue: Reviewer's comments.
R= Reviewer; P = Page; L = Line as they appear in the Original manuscript version
G= General comment followed by numbering

In black: Answers to referees.
P=Page; L=Line; Track change version
*In black and italic: Modification added to text.*

**Comments from the Reviewers:**

**Reviewer #2:**

Synopsis:

The manuscript by Mavrovic et al. conducted permittivity measurements of different soil types with various soil water content using OECP and HydraProbe at frequency of L-band and 50MHz, respectively. Two experiments, fast freeze/thaw transition and slow freeze/thaw transition, were designed. Two soil dielectric model, TD GRMDM and Zhang's model, were driven by the known inputs to simulate the real and imaginary part of soil permittivity. By comparing permittivity measurements between OECP and HydraProbe during freeze/thaw cycles, they demonstrated there are differences of permittivity characteristic between L-band and MHz instruments and suggested the necessities to make proper calibration. By comparing the permittivity measurements and model simulations, they reported the observable discrepancies and highlighted the need for soil dielectric models to take into account the hysteresis effect. Such work is under the research topic to evaluate satellite microwave data products from the in situ permittivity measurements (MHz frequency).

The topic of this manuscript is of interest to the readers of HESS and the measurements can be potentially of importance to the microwave related researches. However, in its current form, the uncertainties regarding the measurements are not detail, which make it hard to judge the validity of the comparison of OECP and HydraProbe measurements. The difference between OECP and HydraProbe measurements is not only from the frequency dependence of permittivity, but also can come from the fact that they are not measuring the same volume of soil samples. As the temperature range of this experiment is large, the temperature dependence of OECP and HydraProbe measurements matters. In addition, the presentation of results is with inaccuracies and can be further explained. Given the current form of the manuscript, I cannot recommend its publication. I expect it suitable for publication in HESS with convinced presentation of measurements and results. Please see below my specific comments.

We made substantial improvement to the manuscript by adding a more explicit presentation of the OECP uncertainties and calibration dependency on temperature. Clarifications were also added on the impact of the temperature sensing volume versus the permittivity sensing volume on the hysteresis amplitude. The Results section was restructured with added explanations on the permittivity measurements. Details were added on the models' equations and assumptions.

Specific comments:

[1] R2, P1, Title and Abstract: I can not see any details about the description of soil dielectric characterization in the Abstract. Please consider either adjust the title or adding the relevant text in Abstract.

The title was adjusted to better reflect the study objectives.

P1, Title: Soil dielectric characterization during freeze-thaw transitions *using L-band coaxial probe and soil moisture probes*

[2] R2, P2, L71: "The high uncertainties in soil permittivity models result from the difficulty in gathering in situ permittivity…" as from my understanding, the uncertainties in soil permittivity models can come from the parameters is not well defined by the in situ permittivity measurements. please clarify this sentence.

The sentence was rewritten to clarify the impact of the lack of reliable microwave permittivity measurements on model parameterization and validation.

P2-3, L74-76: The difficulty in gathering *in situ* permittivity data at microwave frequencies *represents a major hindrance in the parameterization and validation of soil permittivity models, which induces high uncertainties in soil permittivity estimates*.

[3] R2, P3-4, L104 and L130: Section numbers are incorrect.

Corrected.

P4, L108: 2.1.1 Open-Ended Coaxial Probe (OECP)

P4, L134: 2.1.2 HydraProbe

[4] R2, P4, L124: please explain the temperature dependence of OECP measurements. As OECP undergoes a large variation of temperature (e.g., -10ºC to 10ºC), how does OECP perform under such conditions? At which temperature OECP is calibrated? Please make a clarification.

Details were added about the calibration of the OECP in this experiment. Since the OECP is an instrument in development, the calibration of the instrument is undertaken as frequently as possible, although there is a small temperature dependency, it was smaller than the measurement uncertainties. Commercial instruments, such as HydraProbes, are

typically judged to have a calibration stable enough to have confidence in the manufacturer calibration throughout the temperature range.

P7, L220-228: *The OECP was calibrated (see Sect. 2.1.1) in the temperature-controlled chamber at +10ºC. The OECP can operate at a wide range of temperature and was tested to temperature down to -30ºC in the Canadian Arctic (Mavrovic et al., 2020). Beside the OECP, the Planar R54 reflectometer (Copper Mountain Technologies) generating and measuring the electromagnetic waves is graded for [-10 +50] ºC temperature range and the Pasternack coaxial cable joining the OECP and the reflectometer for [-50 +205] ºC temperature range. The OECP calibration displays a slight temperature dependency, where the calibration drift showed a 0.5% increase in permittivity when using a calibration at -15 ºC compare to a calibration at 10 ºC. This calibration drift is small compared to the measurement uncertainties (±3.3% for real permittivity and ±2.5% for imaginary permittivity; Mavrovic et al., 2018).*

[5] R2, P5, Sect. 2.2: please consider presenting the equations used for TD GRMDM and Zhang's model, maybe can put in the appendix. As later you proposed a modification of Zhang's model to consider the hysteresis effect, It is better to present the equations and clear introduce how you make modifications.

Substantial details were added on the Zhang's model and TD GRMDM used for this study. The modifications to Zhang's model that were made to produce fig. 10 were explained in the response to comment R2[14].

P5-6: The order of Sections 2.2.1 and 2.2.2 were interchanged.

P5, L156-179: The model from Zhang et al. (2010) (henceforth Zhang's model) is a semi-empirical soil model for estimating microwave soil permittivity from soil physical characteristics. It is an extension of the semi-empirical mixing dielectric model (SMDM) adapted to frozen soils from Dobson et al. (1985). *Zhang's model is based on dielectric mixing for soil/air/water mixture to estimate soil permittivity at microwave frequencies:*

$$\varepsilon^\alpha = f_s \varepsilon_s^\alpha + f_a \varepsilon_a^\alpha + f_{fw} \varepsilon_{fw}^\alpha + f_{bw} \varepsilon_{bw}^\alpha + f_i \varepsilon_i^\alpha \tag{1}$$

*where $\varepsilon$ is the permittivity of the overall soil mixture, $\alpha$ a constant shape factor (optimized at 0.65 by Zhang et al., 2003), f the fraction of each component in the soil mixture and the subscripts s, a, i, fw and bw refer respectively to solid soils, air, ice, free water and bound water. The approximation of combining free and bound water is made in the model to avoid evaluating the challenging bound water permittivity ($\varepsilon_w$). Also, air contribution to permittivity is negligible ($\varepsilon_a \approx 1$). Zhang's model evaluates the unfrozen water fraction ($f_w$) in soil near the freezing point in order to obtain a continuous transition between the solid and liquid phases of water. An empirical exponential decay function ($f_w = A \cdot |T_{soil}|^{-B}$) is used to estimate the liquid water vs. ice fractions in the freezing soils.* The parameters A and B of the previous function were empirically estimated based on soil types (Zhang et al., 2003). *Solving eq. 1 to obtain an expression for soil mixture permittivity from constant and measurable parameters, Zhang et al. (2010) obtained:*

$$\varepsilon^{\alpha} = 1 + \frac{\rho_b}{\rho_s}(\varepsilon'^{\alpha}_s - 1) + f_w^{\beta}\varepsilon_w^{\alpha} - f_w + f_i\varepsilon_i^{\alpha} - f_i \tag{2}$$

*where $\rho_b$ represents soil bulk density, $\rho_s$ soil specific density and $\beta$ is a parameter that depends on soil composition.* The input parameters required by Zhang's model *to evaluate all variables in eq. 2* include frequency (set at 1.4 GHz for this study), soil moisture (main driver for soil permittivity), temperature, dry bulk density and composition (clay, silt and sand fractions) (Zhang et al., 2003 and 2010; Mironov, 2017).

P6, L182-183: The TD GRMDM is a semi-empirical model that estimates the microwave permittivity of a soil from its physical properties *using a mixing dielectric approach similar to Zhang's model* (Mironov et al., 2010).

P6, L186-189: *The computational implementation of the TD GRMDM used in this experiment was provided by members of the CESBIO team (Centre d'Etudes Spatiales de la Biosphère, Toulouse, France) that worked on the operational product of the SMOS mission which used TD GRMDM as one of its modelling components.*

[6] R2, P6, L193: what is HPP?

The HPP was not used in this experiment, the reference was removed.

P7, L230: HP output signals were logged with a CR800 datalogger (Campbell Scientific, Inc.).

[7] R2, P6, Sect. 3.1.2: Slow freeze/thaw transition Please explain the purpose for this experiment. Please describe the temperature settings and add information about the measuring interval of OECP and HP measurements.

The purpose of the slow freeze/thaw transition is to observe the effect of transition speed on the amplitude of the hysteresis effect. This precision was added along with details on the temperature settings and variable measurement interval. It should be noted that no HP measurements were taken during the slow freeze/thaw transition experiment.

P7, L240-251: To investigate the effect of *a slower freeze/thaw transition on the temperature amplitude of the hysteresis effect,* another experimental setup was created in a Climats EXCAL 1411-HE cold chamber (0.138 m$^3$ volume) at the Laboratoire de l'Intégration du Matériau au Système (Bordeaux, France). Since the soil sample and the Polytetrafluoroethylene (i.e. PTFE or TEFLON) container had smaller volumes, the OECP probe was installed on top of the soil sample with its open end in contact with the soil (Fig. 4). Only OECP permittivity measurements were taken in this experiment since an HP sensor was not available. The objective of this experimental setup was to *experiment* a slow freeze/thaw transition. *Measurements were made to cover a soil temperature range from -20$^o$C to +11.5$^o$C with a variable soil temperature measurement interval to have a finer curve resolution around freezing point.* Permittivity measurements were taken only when the soil temperature equilibrated with the cold

chamber air temperature (± 0.1°C). This method was significantly more time-consuming than the fast transition setup, as a full cycle took several days and required heavy user surveillance.

[8] R2, P6-7, Sect. 3.2: Maybe I have misunderstandings here. How many soil samples were collected and then used in this experiment? Are these soil samples for each site with the same moisture content?

The OBS soil sample consists of one sample in which all probes can properly fit. Two samples were used from the three other sites, one for the OECP and another for the HP. Explanations were added on the reason why two distinct setups were used which should enlighten the reader. All soil samples were conserved as close as possible to their original moisture content as collected on the field. Those moisture content levels can be found in Table 1.

P7, L216-218: *The Fig. 3a and 3b setup discrepancies only reflect the two distinct containers used for soil collection at different sites, both configurations ensured sufficient spacing for undisturbed measurements.*

[9] R2, P7, L221: When is the experiment conducted?

Specified.

P8, L261: The samples were collected January $27^{th}$, 2018.

Fig. 5: *Experiment conducted from February $1^{st}$ to February $7^{th}$, 2018.*

Fig. 6: *Experiment conducted from April $15^{th}$ to April $19^{th}$, 2018.*

Fig. 7: *Experiment conducted from March $29^{th}$ to April $6^{th}$, 2018.*

Fig. 8: *Experiment conducted from April $6^{th}$ to April $15^{th}$, 2018.*

Fig. 11: during a slow freeze/thaw cycle in a *temperature-controlled* chamber environment. *Experiment conducted July $12^{th}$, 2017.*

[10] R2, P7-8, Sect. 4: In this section, Figures 5-8 are presented. While only a general description was presented. Lacking of the characteristic of soil dielectric, the difference among Figures 5-8, the difference between fast and slow freeze/thaw transition measurements.

The Experimental Results section (4) was restructured for clarity. More details were added on the description of Figs. 5-8 and Table 2-3. Some comments were also added on the difference between the fast and slow freeze/thaw transitions.

[revised manuscript text omitted]

[12] R2, P7, L246: "with offsets depending on the soil type" please consider presenting the results more detail.

Details were added in this paragraph along with a reorganization of the flow of ideas. See response to comment R2[10].

[13] R2, P8, L267-268: "both models overestimated the soil permittivity of thawed samples with high water content according to the results of this study." please explain such overestimation.

It is hypothesized that the models' permittivity overestimation is due to an underestimation of bound water fraction or bound water permittivity. Evaluating the plausibility of this hypothesis is out of the scope of this study, further investigation would be required. This hypothesis was added to the Results section.

P10, L327-333: Lastly, both models overestimated the soil permittivity of thawed samples with high water content according to the results of this study (Fig. 5)*, which agrees with results from Bircher et al. (2016b). Further investigation would be required to identify the sources of permittivity overestimation in the models, although it is probable that it comes from the difficulty in uncoupled free and bound water in soil permittivity models. The movement of a fraction of water molecules under the soil surface is hindered by solid soil particles. Those constrained water molecules are described as bound water. Since their ability to align with an electrical field is reduced, the permittivity of bound water is reduced as well (Jones et al., 2002).*

P14, L516-517: *Jones, S., Wraith, J., and Or, D.: Time domain reflectometry measurement principles and applications, Hydrol. Process., 16(1), 141-153, doi: 10.1002/hyp.513, 2002.*

[14] R2, P9, L295: please consider presenting the equations of the modified version of Zhang's model.

As describe in the paragraph, the ice fraction was prescribed around freezing point in the Zhang's model using an exponential function ($\frac{e^x}{e^x+1}$). This is the only modification applied to the Zhang's model.

[15] R2, P9, L296: "consider ice fraction above 0°C" is the artefact or the real conditions? Please make explanations.

This should be considered an artefact rather than real conditions, although this artefact is of relevant importance for soil permittivity applications (see response to comment R2[18]).

P10-11, L355-359: The classic Zhang's model only takes into account ice fraction below 0°C, *this ice fraction should not be interpreted as actual ice at temperature below freezing point but rather as an aggregate of the heterogeneous soil temperature.* Figure 10 demonstrates the hysteresis effect simulated by using a modified version of Zhang's model *that considers ice fraction above and below 0°C.*

The hypothesis referred is the one proposed in the previous paragraph about the correlation between the hysteresis effect and the temperature transition speed. It was clarified to avoid further confusion.

P11, L362-363: *We further tested the hypothesis that the hysteresis effect is correlated with the temperature transition speed* using an OBS soil sample using a slower freeze/thaw transition rate.

[17] R2, P9, L316: please explain how you implement a double "threshold"

Some additional explanation on the double threshold proposition was added.

P11, L376-385: To reproduce the hysteresis effect at freeze/thaw transition, two approaches are possible. An empirical approach could be used by implementing a double threshold *using distinct ice fraction empirical relationships for 1) the freezing and 2) the thawing cycle.* This empirical approach would require determining the freezing/thawing temperature offset independently for each transition type which would depend on liquid water content, textural composition, solute concentration, and the pore pressure of the soil (Daanen et al., 2011). The alternative would be to couple dielectric models with soil physical models that integrate the time evolution of soil physical properties (e.g. CLASSIC model; Melton at al., *2020*). Soil physical models provide an estimate of the ice fraction through time, which is used by dielectric models to estimate soil permittivity. Such coupling should only impact the freeze/thaw transition where ice fraction is a relevant parameter.

P15, L562-565: Melton, J., Arora, V., Wisernig-Cojoc, E., Seiler, C., Fortier, M., Chan, E., and Teckentrup, L.: CLASSIC v1.0: the open-source community successor to the Canadian Land Surface Scheme (CLASS) and the Canadian Terrestrial Ecosystem Model (CTEM) – Part 1: Model framework and site-level performance, Geosci. Model Dev., *13(6), 2825–2850, doi:10.5194/gmd-13-2825-2020, 2020.*

[18] R2, P10, Sect. 6: In the current form, conclusion appears not informative compared to the Abstract. Please consider making modifications, adding more information.

Modifications to the conclusion were made and information were added to the hysteresis effect implications.

P8, L281-282: *The repeatability of the OECP measurements can also be seen as an indicator of the reliability of the measurements.*

P11, L395-397: *Although this phenomenon should be considered as an aggregate of soil temperature heterogeneity rather than actual conditions, it is of relevant interest to study and understand it for all macroscopic to satellite scale applications.*

P11, L387-392: *This study presents soil microwave permittivity measurements during freeze/thaw transitions in* the same frequency range as the SMAP and SMOS satellites, as well as future L-band satellite missions. *The permittivity measurements were taken using a novel open-ended coaxial probe (OECP).* It is shown that lower frequency (MHz) soil permittivity probes can be used to estimate microwave permittivity given proper calibration relative to an L-band probe*, which holds significant potential considering the already widespread operational networks of low frequency soil permittivity probes deployed to measure soil moisture.*

[19] R2, P3, L95: considering change into "Section 2.2 gives an overview of two soil permittivity models"

Specified.

P3, L99-100: …Section 2.2 gives an overview of *two* soil permittivity models used for satellite retrieval…

[20] R2, Fig. 4: please add the plotting scale to indicate the dimensions.

Scale added in Fig. 4.

Fig. 4: Updated in manuscript.

[21] R2, Fig. 10: how is it reproduced? Please indicate the equations, the used parameters.

We updated and clarified the equations. For details on both the equations and parameters used, see response to comments R2[5] and R2[14]

[22] R2, Fig. 11: where are (a) and (b) on the figures?

Fig. 4 and 11 were joined in previous versions of the manuscript. Corrected.

Fig. 11: Real ($\varepsilon$') and imaginary ($\varepsilon$'') permittivity of an organic soil sample from the Old Black Spruce site (collected May 3rd, 2017) during a slow freeze/thaw cycle in a *temperature-controlled* chamber environment *(experiment conducted during July 2017)*.

[23] R2, Tables: Please consider using the consistent format

The tables' format was standardized.

Tables 1, 2 and 3: Updated in manuscript.

[revised manuscript text omitted]
 | 6.0 (±0.7/11.7%) | 7.0 (±0.7/10.0%) | 5 | 4.51 | 1.2 (±0.2/16.7%) | 2.8 (±0.4/14.3%) | 0.5 | 0.055 |

---

## Author Comment (AC3) · 30 Oct 2020

**Revision of Manuscript: "Soil dielectric characterization during freeze-thaw transitions using L-band coaxial probe and soil moisture probes" by Alex Mavrovic** *et al.*

In blue: Reviewer's comments.
R= Reviewer; P = Page; L = Line as they appear in the Original manuscript version
G= General comment followed by numbering

In black: Answers to referees.
P=Page; L=Line; Track change version
*In black and italic: Modification added to text.*

**Comments from the Reviewers:**

**Reviewer #3:**

Synopsis:

The manuscript presents interesting measurements of soil permittivity at L-band during the freeze-thaw cycles. Results are compared with two commonly used models (Mironov's model and Zhang's model) and hysteresis effects are observed especially for the fast freeze/thaw transitions. The reviewer found these experimental results are valuable and suggests to accept it for publication after addressing the following concerns.

Specific comments:

[1] R3, P2, L63-64: Not only for L-band, higher frequencies are also able to retrieve the land- scape freeze/thaw state. (e.g. Zuerndorfer et al., 1990; Judge et al., 1997; and Zhao et al., 2011). And if possible, future measurements could be extended to higher frequencies, which is important to retrieve snow properties and soil properties under the snow. Please refer to: Zuerndorfer, B. W., England, A. W., Dobson, M. C., & Ulaby, F. T. (1990). Mapping freeze/thaw boundaries with SMMR data. Agricultural and Forest Meteorology, 52(1-2), 199-225. Judge, J., Galantowicz, J. F., England, A. W., & Dahl, P. (1997). Freeze/thaw classification for prairie soils using SSM/I radiobrightnesses. IEEE Transactions on Geoscience and Remote Sensing, 35(4), 827-832. Zhao, T., Zhang, L., Jiang, L., Zhao, S., Chai, L., & Jin, R. (2011). A new soil freeze/thaw discriminant algorithm using AMSR-E passive microwave imagery. Hydrological Processes, 25(11), 1704-1716.

The authors acknowledge that the list of references for freeze/thaw soil state retrieval algorithm is long and cover a large range of microwave frequencies. Suggested references added.

P2, L66-68: This allows for the retrieval of the ground state (freeze/thaw) from passive microwave observations (*Zuerndorfer et al., 1990; Judge et al., 1997; Zhao et al., 2011*; Rautiainen et al. 2012; Roy et al., 2015; Derksen et al, 2017).

P14, L519-521: *Judge, J., Galantowicz, J., England, A., and Dahl, P.: Freeze/thaw classification for prairie soils using SSM/I radiobrightnesses, IEEE Trans. Geosci. Remote Sens., 35(4), 827-832, doi:10.1109/36.602525, 1997.*

P17, L687-689: *Zhao, T., Zhang, L., Jiang, L., Zhao, S., Chai, L., and Jin, R.: A new soil freeze/thaw discriminant algorithm using AMSR-E passive microwave imagery, Hydrol. Process., 25(11), 1704-1716, doi:10.1002/hyp.7930, 2011.*

P17, L691-692: *Zuerndorfer, B., England, A., Dobson, M., and Ulaby, F.: Mapping freeze/thaw boundaries with SMMR data, Agr. Forest Meteorol., 52(1-2), 199-225, doi:10.1016/0168-1923(90)90106-G, 1990.*

[2] R3, P4, L125: Would it cause uncertainties of measurement when applying different pressures to the soil with the OECP probe?

Yes, adding pressure to the probe would typically increase permittivity measurements because of the densification of the soil. On the other side, if the proper is not properly in contact with the soil, air gaps would produce artificially low permittivity measurements. There is a proper equilibrium to be found here, which is accomplished by digging a small-scale trench, alike soil moisture probes are generally installed in the field. The OECP is placed horizontally to position the probed volume in the undisturbed soil. The OECP is then fully buried to avoid air gap without risking of applying extra pressure. Precision was added about the installation method.

P6-7, L212-216: The OECP and HP were *horizontally inserted into undisturbed soil* and centered at a depth of 2.5 cm below the soil surface *with sufficient spacing between the probes and the soil samples edges to ensure that the probed volumes are restricted to the limits of the soil samples* (Fig. 3). *Special care was deployed to ensure no air gap was found between the OECP and the undisturbed soil, but without applying extra pressure on the probe.*

[3] R3, P7, L248: How are the data points selected for Figure 9, as there are many measurements as shown from Figure 5 to 8. The challenge is how to well model the soil permittivity during the freeze-thaw transitions, and data points during the freezing/thawing period should be included.

Data points of Fig. 9 are the displayed in Table 2 and 3 which represents stable plateau when the soil is fully frozen or thawed, from -6ºC to -5ºC or from +5ºC to + 6ºC respectively. Fig. 9 allows for a comparison of measurements and modelling of soil permittivity away from the freeze/thaw transition, therefore values around freezing point and the hysteresis effect are avoided. Including data near the hysteresis would bias the results for the desired comparison aimed at with Fig. 9.

[4] R3, Fig. 9: please specify those numbers are for RMSE in the figure.

RMSE added in Fig. 9.

Fig. 9: Updated in manuscript.

[5] R3, P9, L290: It is very interesting that the hysteresis effects were observed during the permittivity measurement. As mentioned below by the authors, an empirical approach could be used by implementing a double threshold. It is suggested to do so to discuss the improvement of the model performance compared with results from Figure 9.

Some additional explanation on the double threshold proposition was added. As for the implantation of such proposition in current soil permittivity models, this work will be reserved for future studies.

P11, L376-385: To reproduce the hysteresis effect at freeze/thaw transition, two approaches are possible. An empirical approach could be used by implementing a double threshold *using distinct ice fraction empirical relationships for 1) the freezing and 2) the thawing cycle.* This empirical approach would require determining the freezing/thawing temperature offset independently for each transition type which would depend on liquid water content, textural composition, solute concentration, and the pore pressure of the soil (Daanen et al., 2011). The alternative would be to couple dielectric models with soil physical models that integrate the time evolution of soil physical properties (e.g. CLASSIC model; Melton at al., *2020*). Soil physical models provide an estimate of the ice fraction through time, which is used by dielectric models to estimate soil permittivity. Such coupling should only impact the freeze/thaw transition where ice fraction is a relevant parameter.

[revised manuscript text omitted]
 | 6.0 (±0.7/11.7%) | 7.0 (±0.7/10.0%) | 5 | 4.51 | 1.2 (±0.2/16.7%) | 2.8 (±0.4/14.3%) | 0.5 | 0.055 |

---

## Referee Report (RR1)

Thank the authors a lot for their dedicated revisions, elaborating the potential measurement uncertainties and adding more explanations about the experiment, used models, and results, which addressed most of my concerns. I only have some minor comments for consideration.

Line 246: Please check the appropriate use of "experiment"?

Line 249: "Permittivity measurements were taken only when the soil temperature equilibrated with the cold chamber air temperature ($\pm$ 0.1°C)."

Please explain more about the cold chamber air temperature. The cold chamber air temperature is measured by sensors or the temperature settings of the cold chamber. If for the later option, what is the range of the temperature fluctuations when you set the cold chamber to a specific temperature value?

Section 2.2.1, and Line 357-359, Line 380:

The treatment of ice fraction in Zhang's model is still not mathematically clear to me.

In its original form (Equations 1 &2), the liquid water fraction is calculated as the exponential function of soil temperature. Then how is the ice fraction calculated?

In the updated Zhang's model with consideration of the hysteresis effect, the ice fraction is added as an exponential function ($\frac{e^x}{e^x+1}$). What is "$x$" represented for? How the ice fraction is differed for the freezing and thawing cycles?

The definition of ice fraction as "this ice fraction should not be interpreted as actual ice at temperature below freezing point but rather as an aggregate of the heterogeneous soil temperature" is only for the thawing cycle or for both the freezing and thawing cycles? The cases where "this definition of ice fraction" is used should be clearly indicated.

The concept "freezing/thawing temperature offset" needs more explanation (Line 380).

Section 2.2.1, Equations 1 &2, Table 1:

Although the symbols are well defined, I think it is better to keep the symbols ($\theta_V, \theta_G, \rho_d$) in Table 1 consistent with Equations 1&2.

---

## Referee Report (RR2)

Comments reviewer 1 (Jan Hofste) on revised manuscript: "Soil dielectric characterization during freeze-thaw transitions using L-band coaxial probe and soil moisture probes", Alex Mavrovic, Renota Pardo Lara, Aaron Berg, François Demontoux, Alain Royer, and Alexandre Roy, HESS, 2020

Date: 2020 12 16

[1] Accept author reply and corresponding manuscript revisions.

[2] Author reply clear, but revision in manuscript not yet sufficient. Mention in text that OECP measurements were only performed at one position of OBS sample because only one OECP was available (applies to OBS sample). Mention also that the other three samples were too small to allow for measuring at multiple positions (at same depth of course), doing so would disturb the samples because they would then have holes in them. (As author explains in response to comment [4].)
The low number of OECP sampling positions is, unfortunately, a shortcoming of the experiment. The authors should be honest about this.

Finally, I disagree with the sentence "The repeatability of the measurements gives us confidence that the experimental protocol is robust." Measurements at same positions are indeed alike, and thus are repeatable (albeit there is still some variation..) but this does not have to mean that the retrieved epsilon values are accurate.

[3] Author reply clear, but revision in manuscript not yet sufficient.

I mis the sentence found in the original manuscript (line 244) explaining the hysteresis: "hysteresis should be expected because of the latent heat of fusion of water". Line 287 of new manuscript not necessary: you don't need to give the definition of hysteresis. The hysteresis-amplification is explained better now.

[4] Accept author reply and corresponding manuscript revisions.

[5] Accept author reply and corresponding manuscript revisions.

[6] Author reply and corresponding manuscript revision not yet sufficient. You mention in the explanation and in the manuscript (line 335) the (hysteresis) trends are similar between the permittivity measurements. This statement should be more specific and quantified. Are measurements similar between thaw/freeze cycles?, or between HP positions?, or between different soil samples? Based on the theoretical curve in figure 10 you can define quantities such as $\Delta T$, $\Delta \varepsilon'$, maximum steepness of the slope, and the positions where the slopes are steepest. These quantities you then apply to the various measurements. Based on that you can then also make statements on for example the repeatability (see also comment [27]).

[7] Accept author reply and corresponding manuscript revisions.

[8] Author reply and corresponding manuscript revision not yet sufficient. Don't you mean hysteresis amplification -effect? Because the hysteresis itself is known to be present regardless of any probing volume.

[9] Accept author reply and corresponding manuscript revisions.

[10] Accept author reply and corresponding manuscript revisions.

[11] Accept author reply and corresponding manuscript revisions.

[12] Accept author reply and corresponding manuscript revisions.

[13] Accept author reply and corresponding manuscript revisions.

[14] Accept author reply and corresponding manuscript revisions.

[15] Accept author reply and corresponding manuscript revisions.

[16] Accept author reply and corresponding manuscript revisions.

[17] Accept author reply and corresponding manuscript revisions.

[18] Author reply and corresponding manuscript revision not yet sufficient. Line 122 should be: "The sensing depth is inversely proportional to the medium's permittivity and proportional to the magnitude of the electric field generated by the reflectometer, which provides a constant power output of 10 dBm (Fig. 1b). ". This is also what is shown in Figure 2.

[19] See comment [18].

[20] Accept author reply and corresponding manuscript revisions.

[21] Accept author reply and corresponding manuscript revisions.

[22] Accept author reply and corresponding manuscript revisions.

[23] Accept author reply and corresponding manuscript revisions.

[24] Accept author reply and corresponding manuscript revisions.

[25] See comment [6]

[26] Accept author reply and corresponding manuscript revisions.

[27] Author reply and corresponding manuscript revision not yet sufficient. I disagree with added lines " The repeatability of the OECP measurements can also be seen as an indicator of the reliability of the measurements" (281-283) in revised manuscript. Measurements can have a high repeatability, yet be inaccurate at the same time. You can make quantitative statements on the repeatability, but you assume the epsilon values you measure are accurate based on the calibration of your probe and on whether the sample containers or sample edges don't influence the measurement.

[28] Accept author reply and corresponding manuscript revisions.

[29] Accept author reply and corresponding manuscript revisions.

---

## Author Response (AR2)

**Revision of Manuscript: "Soil dielectric characterization during freeze-thaw transitions using L-band coaxial probe and soil moisture probes" by Alex Mavrovic *et al.**

In blue: Reviewer's comments.
[ ] = Numbering (coherent with the first round of reviewer's comments), R= Reviewer; P = Page; L = Line as they appear in the Original manuscript version
G= General comment followed by numbering

In black: Answers to referees.
P=Page; L=Line; Track change version
*In black and italic: Modification added to text.*

**Comments from the Reviewers:**

**Reviewer #1:**

Synopsis:

Comments reviewer 1 (Jan Hofste) on revised manuscript: "Soil dielectric characterization during freeze-thaw transitions using L-band coaxial probe and soil moisture probes", Alex Mavrovic, Renota Pardo Lara, Aaron Berg, François Demontoux, Alain Royer, and Alexandre Roy, HESS, 2020

We made improvement to the manuscript by adding details on the experimental setup, the accuracy/precision of the OECP and our experimental setup, and the similarities between the OECP and HP measurements, along with clarifications where the reviewer's comments suggested doing so.

Specific comments:

[2] R1, G1: General remark on the samples and measurement setup. With the OBS sample HP measurements were taken at three positions. As Figure 5 shows the measured responses at these three positions varies. Why were there not also measurements at multiple positions for the OECP with the OBS sample? And why were the other 3 samples not also measured with the HP (and the OECP) at multiple positions? Was this because the OBS sample was expected to be less homogeneous due to the organic content? And why only one sample per soil type was measured? The choices the authors made in this regard should be explained in the text, even if simply for practical reasons.

The OECP is a promising instrument currently developed by the Université de Trois-Rivières and Université de Sherbrooke. Only one OECP was available for the experiment. Logistics is the primary reason for the difference in setup between the OBS soil samples and the others, the soil sample collections were not made in the same type of container for all sites. Even if the cylinder samples are smaller in size, the probed were

properly positioned to ensure reliable measurements (see previous response to comment R1[1]). The repeatability of the measurements gives us confidence that the experimental protocol is robust. The explanation for the two distinct setups was added.

P7, L216-218: *The Fig. 3a and 3b setup discrepancies only reflect the two distinct containers used for soil collection at different sites, both configurations ensured sufficient spacing for undisturbed measurements.*

Author reply clear, but revision in manuscript not yet sufficient. Mention in text that OECP measurements were only performed at one position of OBS sample because only one OECP was available (applies to OBS sample). Mention also that the other three samples were too small to allow for measuring at multiple positions (at same depth of course), doing so would disturb the samples because they would then have holes in them. (As author explains in response to comment [4].) The low number of OECP sampling positions is, unfortunately, a shortcoming of the experiment. The authors should be honest about this.

Revisions in the manuscript were added to reflect more clearly the author reply to the reviewer's comment.

P7, L218-222: The Fig. 3a and 3b setup discrepancies only reflect the two distinct containers used for soil collection at different sites, both configurations ensured sufficient spacing for undisturbed measurements. *OECP measurements were performed at only one position in each experiment because only one OECP was available. The setup of Fig. 3b only includes one HP position because of containers' size limitation.*

Finally, I disagree with the sentence "The repeatability of the measurements gives us confidence that the experimental protocol is robust." Measurements at same positions are indeed alike, and thus are repeatable (albeit there is still some variation..) but this does not have to mean that the retrieved epsilon values are accurate.

It was clarified that the cited previous work already provides estimates on the OECP accuracy and precision with quantified uncertainties. The repeatability of the measurements gives us a certain confidence that the experimental setup is suitable to obtain reliable soil permittivity measurements during freeze/thaw transitions.

P9, L285-291: *Previous work already shown that the OECP is a reliable instrument to measure a medium's permittivity such as tree trunks (Mavrovic et al., 2018), leaves (Holtzman et al., accepted) and snow (Mavrovic et al., 2020). The OECP displays uncertainties under 3.3% and 2.5% for real and imaginary permittivity respectively when tested on reference materials (Mavrovic et al., 2018). In this study,* the repeatability of the OECP measurements *through several freeze/thaw cycles* can also be seen as an indicator of the reliability of the *experimental setup to measure soil permittivity during freeze/thaw transitions with the OECP and HP.*

P14, L518-521: *Holtzman, N., Anderegg, L., Kraatz, S., Mavrovic, A., Sonnentag, O., Pappas, C., Cosh, M., Langlois, A., Lakhankar, T., Tesser, D., Steiner, N., Colliander, A., Roy, A., Konings, A.: L-band vegetation optical depth as an indicator of plant water potential in a temperate deciduous forest stand, Biogeosciences, doi: 10.5194/bg-2020-373, accepted.*

[3] R1, P7, L243: The amplification of the hysteresis -effect by the setup, is it possible to explain this in the text with a few sentences? You refer to this hysteresis amplification later on, it would be better if the reader could find an explanation for this effect in this manuscript rather than somewhere else (the reference). You can of course leave the reference.

Further explanations and reference were added.

P8-9, L286-291: Hysteresis effects can be observed between the freezing and thawing cycles in Figs. 5 through 8*, i.e. a different behavior of permittivity variation depending on whether the ground freezes or thaws*. Although hysteresis *is reported in soil freezing studies, this effect was amplified by the temperature transition speed and differences in the sensing volume for temperature and permittivity observations (Pardo Lara et al., 2020)*. Fig. 11 shows a slow freeze/thaw transition displaying a hysteresis effect of diminished amplitude, but still noticeable.

P9-10, L321-323: Even if amplified by the experimental setup, the hysteresis effect between the freezing and thawing cycles is not simulated by any model since *they do not include the evolution of soil properties in time*.

I mis the sentence found in the original manuscript (line 244) explaining the hysteresis: "hysteresis should be expected because of the latent heat of fusion of water". Line 287 of new manuscript not necessary: you don't need to give the definition of hysteresis. The hysteresis-amplification is explained better now.

The unnecessary hysteresis definition was removed. The hysteresis amplification explanation based on the difference in permittivity and temperature sensing volume replaced the latent heat explanation because it was deemed more plausible.

P9, L295-298: Hysteresis effects can be observed between the freezing and thawing cycles in Figs. 5 through 8. Although hysteresis is reported in soil freezing studies, this effect was amplified by the temperature transition speed and differences in the sensing volume for temperature and permittivity observations (Pardo Lara et al., 2020 *and in review*).

P16, L629-631: Pardo Lara, R., Berg, A., Warland, J., Parkin, G.: Implications of measurement metrics on soil freezing curves: A simulation of freeze-thaw hysteresis, Hydrol. Process., doi:10.22541/au.160466100.02966301/v1, in review.

[6] R1, P8, L270-272: Based on the Figures 5 - 8 I find the freeze/thaw transitions not similar. Can the differences of the OECP and HP measurements be explained by the difference in probing volume? Also you mention that the main difference between the OECP and HP measurements are the epsilon values at the end of the cycle, at the "stable plateaus" as you call it. But isn't the hysteresis just as important? Perhaps if a found calibration equation for a given soil is applied to the HP results the freeze/thaw hysteresis is more like that of the OECP?

It is correct that the difference in freeze/thaw transition steepness could be explained by the difference in probing volume. The authors share the same point of view that this is probably the main explanation and it is put forward in the Experimental Results section (4.1).

The fully frozen/thawed values comparison between measurements and models consist of the strongest differences observed in this study. The hysteresis is of equal importance, but the trends are similar between the permittivity measurements. This is to say that the hysteresis effect occurs at very similar temperatures.

It is typical to use soil specific calibration equation to produce soil moisture estimates from HP raw permittivity measurements. However, the HP instrument does not allow for customized calibration equation to compute permittivity from raw reflection coefficients. We clarify few points in the manuscript:

P9, L 310-315: It can also be observed that the freeze/thaw transition measurements are *steeper* with the *OECP* than the HP. This is probably due to the HP's larger probed volume. Since the instruments measure an average permittivity for the whole probed volume, a larger probed volume will record a *more extended* freeze/thaw transition because of the longer time required for the freezing/thawing fronts to penetrate the depth of volume probed. *Since the freezing/thawing front is mostly vertically oriented, it is the difference in probes' sensing diameter that causes the difference in transition steepness.*

Author reply and corresponding manuscript revision not yet sufficient. You mention in the explanation and in the manuscript (line 335) the (hysteresis) trends are similar between the permittivity measurements. This statement should be more specific and quantified. Are measurements similar between thaw/freeze cycles?, or between HP positions?, or between different soil samples? Based on the theoretical curve in figure 10 you can define quantities such as $\Delta T$, $\Delta \varepsilon'$, maximum steepness of the slope, and the positions where the slopes are steepest. These quantities you then apply to the various measurements. Based on that you can then also make statements on for example the repeatability (see also comment [27]).

Similarities between permittivities were detailed and quantified.

P10, L343-347: *The soil temperature offsets from water freezing point are consistent between the OECP and HP measurements for both the freezing and thawing transitions. The difference is ranging from -1.00 to +0.83 ºC when evaluating the soil temperature*

*offset at maximum transition rate (Tables S1 and S2).* The main difference between the permittivity measured at microwave and MHz frequencies appears to be *a permittivity* offset *and the temperature span of the freeze/transition* dependent on the soil type.

Tables S1 and S2: Added in supplementary material.

[8] R1, P9, L300: What hypothesis do the authors refer to?

The hypothesis referred is the one proposed in the previous paragraph about the correlation between the hysteresis effect and the temperature transition speed. It was clarified to avoid further confusion.

P11, L368-369: *We further tested the hypothesis that the hysteresis effect is correlated with the temperature transition speed* using an OBS soil sample using a slower freeze/thaw transition rate.

Author reply and corresponding manuscript revision not yet sufficient. Don't you mean hysteresis amplification -effect? Because the hysteresis itself is known to be present regardless of any probing volume.

It was clarified that the hysteresis amplification is specifically referred in the sentence.

P11, L373-374: We further tested the hypothesis that the hysteresis *amplitude* is correlated with the temperature transition speed using an OBS soil sample *with* a slower freeze/thaw transition rate.

[18] R1, P4, L116: "The penetration depth of the… " This sentence is too vague for my taste. I propose something like: "The sensing depth of the OECP is the maximum depth at which the medium is polarized due to the incident electric field, and as such contributes to the reflection of the EM wave backwards into the coax."

Sentence reworked.

P4, L120-127: The *sensing* depth of the OECP *is* defined as the maximal depth at which a medium is *polarized due to the incident electric field, and as such contributes to the electromagnetic wave reflection. The sensing depth is proportional to the medium's permittivity and the magnitude of the electric field generated by the reflectometer*, which *displays a constant power output of 10 dBm* (Fig. 1b). The OECP typical *sensing* depth approaches 1 cm under dry soil conditions and the cylindrical probed volume is about 3.5 cm wide in diameter (Figure 2). Under wet soil conditions, the *sensing* depth shrinks down to 0.4 cm.

Author reply and corresponding manuscript revision not yet sufficient. Line 122 should be: "The sensing depth is inversely proportional to the medium's permittivity and proportional to the magnitude of the electric field generated by the reflectometer, which

provides a constant power output of 10 dBm (Fig. 1b). ". This is also what is shown in Figure 2.

Clarifications added.

P4, L122-124: The sensing depth is *inversely* proportional to the medium's permittivity and *proportional to* the magnitude of the electric field generated by the reflectometer, which displays a constant power output of 10 dBm (Fig. 1b).

[19] R1, P4, L118: "The magnitude of this effective electric… " the effective electric field has not been defined or explained previously. I assume you refer the resulting electric field in the medium? Which is the sum of the original electric field coming from the coax E0, which polarizes (rotates and or translates) the molecules and the electric field produced by the rotated or displaced molecules themselves Ed. Latter counters E0, which counters Ed, which counters E0 etc. You end up with a resulting electric field E, which is actually lower in magnitude for a higher epsilon.

The effective electrical field refers here to the extent of the electrical field influencing the reflection coefficient measurements. The use of this term here seems confusing and not necessary. Therefore, it was removed and the sensing depth was directly referenced.

P4, L122-124: *The sensing depth is proportional to the medium's permittivity and the magnitude of the electric field generated by the reflectometer*, which *displays a constant power output of 10 dBm* (Fig. 1b).

See comments [18].

See response to comment R1[18].

[25] R1, P7, L243 – 246. Authors state that trends of OECP and HP are "very similar" and the fully frozen/thawed epsilon values are "also similar". I disagree with this description. Judging from Figures 5 - 8 there are significant differences. These differences and explanations for their causes are discussed further down in the text.

The similarity mentioned here was meant to point at the closer similarities between OECP and HP measurements than the model estimates. Since the model results are discussed in the next section, the sentence was reformulated to remove this mention.

P9, L296-299: The HP measurements show trends *in agreement with* the OECP measurements during freeze/thaw transitions, especially for the real permittivity*, although the fully* frozen and thawed permittivity values *display soil type dependent offsets* between the OECP and HP measurements (Tables 2 and 3).

See comment [6]

See response to comment R1[6].

[27] R1, P10, L325 - 327: The question whether the OECP correctly measures the epsilon in not shown in this manuscript. It is implied by your earlier work, see also comment [10].

The reliability of OECP measurements are not thoughtfully investigated in this study, although confidence in the reliability of the measurements can be inferred from the repeatability through freeze/thaw cycles. The conclusion was adapted to shift focus from OECP reliability to soil permittivity results and a hint to OECP measurements repeatability was added in the Results section.

P8, L281-282: *The repeatability of the OECP measurements can also be seen as an indicator of the reliability of the measurements.*

P11, L387-392: *This study presents soil microwave permittivity measurements during freeze/thaw transitions in* the same frequency range as the SMAP and SMOS satellites, as well as future L-band satellite missions. *The permittivity measurements were taken using a novel open-ended coaxial probe (OECP).* It is shown that lower frequency (MHz) soil permittivity probes can be used to estimate microwave permittivity given proper calibration relative to an L-band probe*, which holds significant potential considering the already widespread operational networks of low frequency soil permittivity probes deployed to measure soil moisture.*

Author reply and corresponding manuscript revision not yet sufficient. I disagree with added lines " The repeatability of the OECP measurements can also be seen as an indicator of the reliability of the measurements" (281-283) in revised manuscript. Measurements can have a high repeatability, yet be inaccurate at the same time. You can make quantitative statements on the repeatability, but you assume the epsilon values you measure are accurate based on the calibration of your probe and on whether the sample containers or sample edges don't influence the measurement.

See response to comment R1[2].

**Reviewer #2:**

Synopsis:

Thank the authors a lot for their dedicated revisions, elaborating the potential measurement uncertainties and adding more explanations about the experiment, used models, and results, which addressed most of my concerns. I only have some minor comments for consideration.

We made improvement to the manuscript by adding clarifications where the reviewer suggested doing so, namely on the Zhang's model ice fraction estimation and on the slow freeze/thaw transition experiment.

Specific comments:

[30] R1, P7, L246: Please check the appropriate use of "experiment"?

A more suitable formulation replaced ''experiment''.

P7, L249-250: The objective of this experimental setup was to *undergo* a slow freeze/thaw transition.

[31] R1, P7, L249: "Permittivity measurements were taken only when the soil temperature equilibrated with the cold chamber air temperature (± 0.1°C)." Please explain more about the cold chamber air temperature. The cold chamber air temperature is measured by sensors or the temperature settings of the cold chamber. If for the later option, what is the range of the temperature fluctuations when you set the cold chamber to a specific temperature value?

The cold chamber temperature was measured by the Climats EXCAL 1411-HE cold chamber temperature sensors and permittivity measurements were taken when the cold chamber air temperature stabilized and the fluctuations between the cold chamber air temperature and the soil sample temperature were under ± 0.1°C. It was clarified in the manuscript.

P8, L252-254: Permittivity measurements were taken only when *the cold chamber air temperature measurements stabilized and the fluctuations between the air and soil temperature were under* ± 0.1°C.

[32] R1, P5, Sect. 2.2.1 and P10-11, L357-359 and P11, L380: The treatment of ice fraction in Zhang's model is still not mathematically clear to me.

In its original form (Equations 1 &2), the liquid water fraction is calculated as the exponential function of soil temperature. Then how is the ice fraction calculated? In the updated Zhang's model with consideration of the hysteresis effect, the ice fraction is added as an exponential function ($\frac{e^x}{e^x+1}$). What is "x" represented for? How the ice fraction is differed for the freezing and thawing cycles?

Precisions were added on the ice fraction calculation for the original Zhang's model. It was also specified that the ''x'' in the exponential function refers to soil temperature and that a temperature offset was used to reproduce the hysteresis effect.

P5, L170-172: An empirical exponential decay function ($f_w = A \cdot |T_{soil}|^{-B}$) is used to estimate the liquid water *fraction in the freezing soils, the ice fraction is determined from the liquid water fraction and the total amount of water in the soil.*

P11, L369-370: This ice fraction was prescribed following an exponential function $(\frac{e^{T_{sol}}}{e^{T_{sol}}+1})$ around the freezing point *with a ±0.5ºC temperature offset for the freezing and thawing cycles*.

The definition of ice fraction as "this ice fraction should not be interpreted as actual ice at temperature below freezing point but rather as an aggregate of the heterogeneous soil temperature" is only for the thawing cycle or for both the freezing and thawing cycles?

The correct wording should have been ''liquid water'', it was corrected.

P12, L365-367: The classic Zhang's model only takes into account ice fraction below 0°C, *the resulting liquid water* fraction should not be interpreted as actual *liquid water* at temperatures below freezing point but rather as an aggregate of the heterogeneous soil temperature.

The cases where "this definition of ice fraction" is used should be clearly indicated. The concept "freezing/thawing temperature offset" needs more explanation (Line 380).

Explanations were added on the temperature offset proposition.

P11-12, L391-394: This empirical approach would require determining *independently for each transition type the freezing/thawing hysteresis amplitude as a temperature offset between the state transition and 0ºC. This* would depend on liquid water content, textural composition, solute concentration, and the pore pressure of the soil (Daanen et al., 2011).

[33] R1, P5, Sect. 2.2.1 and P5, Eq. 1-2 and P24, Table 1: Although the symbols are well defined, I think it is better to keep the symbols ($\theta_V$, $\theta_G$, $\rho_d$) in Table 1 consistent with Equations 1&2.

The symbols were standardized through the manuscript based on the symbol defined in Eq. 1 and 2.

P24, L809, Table 1: $f_i(V)$ and $f_i(G)$ stands for volumetric and gravimetric liquid water content, respectively.

P24, Table 1: Updated in manuscript.